# Advantage Actor-Critic Training Framework Leveraging Lookahead Rewards for Automatic Question Generation

## Abstract

Existing approaches in Automatic Question Generation (AQG) train sequence-to-sequence (seq2seq) models to generate questions from input passages and answers using the teacher-forcing algorithm, a supervised learning method, resulting in exposure bias and training-testing evaluation measure mismatch. Several works have also attempted to train seq2seq models for AQG using reinforcement learning, leveraging Monte-Carlo return-based policy gradient (PG) methods like REINFORCE with baseline. However, such Monte-Carlo return-based PG methods depend on sentence-level rewards, which limits the training to sparse and high-variance global reward signals. Temporal difference learning (TD)-based Actor-Critic methods can provide finer-grained training signals for solving text-generation tasks by leveraging subsequence-level information. However, only a few works have explored the Actor-Critic methods for text generation because it becomes an additional challenge to train the seq2seq models steadily using such TD methods. Another severe issue is the vocabulary size-related intractable action space bottleneck inherent in all natural language generation (NLG) tasks. This work proposes an Advantage Actor-Critic training framework to train seq2seq models for AQG, which uses sub-sequence level information to train the models efficiently and stably. The proposed training framework also addresses the problems of exposure bias, evaluation measure mismatch and global rewards by facilitating the autoregressive token generation, BLEU-based task optimization and question prefix-based Critic signals and provides a workaround for the intractable action space bottleneck by leveraging relevant ideas from existing supervised learning and reinforcement learning literature. The training framework uses an off-policy approach for training the Critic, which prevents the Critic from overfitting the highly correlated on-policy training samples. The off-policy Critic training also uses an explicit division of high-reward and low-reward experiences, which provides additional improvement to the training process. In this work, we conduct experiments on multiple datasets from QG-Bench to show how the different components of our proposed Advantage Actor-Critic training framework work together to improve the quality of the questions generated by the seq2seq models by including necessary contextual information and ensuring that the generated questions have a high degree of surface-level similarity with the ground truth.

## 1 Introduction

Automatic Question Generation (AQG) is the task of automatically generating questions from natural language text. It is a challenging task that integrates the challenges of Natural Language Understanding (NLU) and Natural Language Generation (NLG). An AQG system aims to generate grammatically correct and contextually relevant questions. As a result, such a system requires an in-depth understanding of the input text and the ability to reason over relevant contexts. Over the years, there has been a growing interest in AQG, and it has found various applications. For example, AQG systems have applications in the educational domain, where such systems create reading comprehension materials for learners (Heilman & Smith, 2010) or are components in adaptive, intelligent tutoring systems (Lindberg et al., 2013). Such systems also have significant business use; they can be crucial components of conversational AI systems (Mostafazadeh et al.,

2016) and help improve the user experience in such conversations. AQG systems also help improve the state-of-the-art in the wider Natural Language Processing (NLP) research domain. Such systems can create large-scale question-answering datasets (Rajpurkar et al. (2016), Nguyen et al. (2016)) for training models for related tasks like Question-Answering (QA).

Recent work in the AQG research focuses more on neural network-based approaches (Du et al. (2017), Yuan et al. (2017), Song et al. (2018), Zhou et al. (2017), Zhao et al. (2018)) where we train a neural network in an end-to-end manner to generate natural language questions from the input text. The neural networks are sequence-to-sequence (seq2seq) models composed of an encoder and a decoder. The encoder is responsible for identifying the essential topics and creating an intermediate representation of input text such that the decoder can generate the appropriate question. In this way, the seq2seq models jointly perform the task of content selection and question construction. This way of question generation is very similar to how such seq2seq models perform other sequence generation tasks like Machine Translation and Text Summarization. However, one crucial difference between AQG and such sequence generation tasks is that we can generate multiple questions from the same content depending on the question's intent. For example, a paragraph on the *Taj Mahal* having Taj Mahal and 1631 as the selected content can lead to either a *When*-type question (*e.g., When was the construction of the Taj Mahal commissioned?*) or a *Who*-type question (*e.g., Who commissioned the construction of the Taj Mahal?*).

In most works using seq2seq models for AQG, people typically train the seq2seq models in the supervised learning setup using the teacher-forcing algorithm or the reinforcing learning setup using Monte Carlo return-based policy gradient methods like Reinforce with baseline. Although quite effective, these methods have many limitations:

- *Exposure bias:* When leveraging the teacher-forcing to train seq2seq models for generating questions, we observe that the model fails to generate good-quality questions from the test data despite the noticeable improvement in the model's performance on the training data. We refer to this as *exposure bias* because the decoder is never exposed to tokens sampled from its distribution and is biased towards the ground-truth tokens.

- *Evaluation measure mismatch:* The supervised learning framework also needs to address the mismatch between evaluation measures used for evaluating the performance of the seq2seq model during training and testing. We use cross-entropy loss during the training stage but discrete, non-differentiable metrics like BLEU during testing. As a result, there needs to be a better match between the training and testing evaluation measures.

- *Global reward problem:* Policy gradient methods like REINFORCE are Monte Carlo return-based methods and use complete sequences to compute sentence-level global rewards and provide training signals to aid the seq2seq model in the question generation process. Since we can calculate the sentence-level rewards only after the generation of the complete question, we set the rewards for the intermediate steps to zero, which hinders our ability to update the seq2seq model's parameters efficiently because we have to rely on sparse reward sequences for training. Additionally, when we use global rewards to aid the seq2seq model training, we have to deal with the problem of high variance in the gradient estimate because the gradients obtained from Monte Carlo return-based policy gradient methods can have high variance, leading to unstable training and slow convergence.

- *Intractable action space bottleneck:* In natural language generation tasks such as AQG, the seq2seq model outputs a token from a vast vocabulary at every decoding step. In the reinforcement learning setup, the tokens generated by the seq2seq model at every decoding step are the actions. Therefore, the action space in such settings is the large vocabulary from which the model selects the tokens. Since this action space is discrete and vast, the RL agent often needs excessive training before converging to a remotely favourable policy or even fails to converge to an appropriate policy. As a result, the intractable action space bottleneck is a severe issue we must deal with appropriately before we can benefit from the Rl-based training frameworks.

Due to the aforesaid limitations, seq2seq models trained using the existing supervised learning and policy gradient-based methods often generate sub-optimal questions. Table 1 gives examples of some output ques-

Table 1: Examples of some output questions generated using teacher-forcing-based supervised learning, policy gradient (PG)-based Self-Critic Sequence Training (SCST), and Advantage Actor-Critic training. For reporting, we only show the sentence fragment from the input passage that contains the answer (coloured in red).

| Example 1 | |
|---|---|
| **Paragraph:** | *Political Economy introduced questions of history and colonialism to ahistorical anthropological theories of social structure and culture. ...* |
| **Target:** | What did Political Economy introduce questions of to theories of social structure and culture? |
| **Supervised Learning:** | Political Economy introduced questions of what? |
| **PG-based SCST:** | Political Economy introduced questions of what? |
| **Advantage Actor-Critic:** | What questions did Political Economy introduce to ahistorical anthropological theories of social structure and culture? |
| Example 2 | |
| **Paragraph:** | *... The Recording Industry Association of America recognized her as the Top Certified Artist in America during the 2000s decade. ...* |
| **Target:** | In which decade did the Recording Industry Association of America recognize Beyonce as the The Top Certified Artist? |
| **Supervised Learning:** | When was Beyoncé recognized as the Top Certified Artist in America? |
| **PG-based SCST:** | When was Beyoncé recognized as the Top Certified Artist in America? |
| **Advantage Actor-Critic** | When did the Recording Industry Association of America recognize Beyoncé as the Top Certified Artist in America? |

tions generated using teacher-forcing-based supervised learning and policy gradient (PG)-based Self-Critic Sequence Training (SCST). These examples show that the questions generated using supervised learning and SCST fail to capture all relevant information.

In this work, we focus on the abovementioned limitations and leverage deep reinforcement learning techniques to propose a stable training framework to aid seq2seq models in the AQG task. Accordingly, we define our problem statement as follows:

*Given a natural language text input comprised of a passage and an answer, use an Actor-Critic training framework to efficiently train seq2seq models for AQG and generate grammatically correct and contextually relevant questions.*

The primary contributions of our work are as follows:

- *Actor-Critic training framework to train seq2seq models for AQG:* In contrast to existing research that primarily uses policy gradient-based approaches to train seq2seq models for AQG, we propose two alternative Actor-Critic method-based training strategies that overcome the PG-based methods' limitations. We refer to the seq2seq model as the Actor in our proposed Actor-Critic frameworks. We use an additional neural network called Critic to generate dense training signals to guide the Actor in the autoregressive question-generation process. The training signals from the Critic help the Actor generate the questions autoregressively using tokens from the Actor's distribution, thereby addressing the exposure bias problem. These signals also help address the evaluation measure mismatch problem by optimizing the Actor towards some specific non-differentiable evaluation metric. The Actor-Critic training also allows us to leverage a learnt signal reinforced with subsequence-level or per-step information to drive the question-generation process.

- *Lookahead reward function to generate dense targets for training the Critic:* Directly using the sentence-level global reward computed at the end of the question-generation process to aid the Actor training is inefficient because the intermediate decoding steps get zero rewards, leading to a sparse reward sequence. Therefore, we use training signals from the Critic to help the Actor generate the questions autoregressively. However, the global rewards obtained using the sentence-level reward function suffer from the same impediment of sparse training targets for the Critic. Therefore, as an alternative, we define a per-step lookahead reward function which generates a scalar lookahead reward upon each addition of a new token and provides dense targets for training the Critic. We use BLEU, a surface-level similarity metric, to define the lookahead reward function to incentivize the Actor to generate questions with significant structural similarity with the ground-truth questions, improving overall fluency.

- *Probability-based action pruning for efficient exploration of vocabulary space:* We focus extensively on the intractable action space problem, a severe bottleneck in applying RL-based techniques for sequence generation tasks like AQG and provide effectual strategies to address the issue. We propose an alternative exploration strategy to the usual softmax sampling-based exploration, which allows the seq2seq model to focus only on a subset of the vast vocabulary space relevant to the question generation process at each decoding step. We argue that this alternative exploration strategy is a better practical approach which helps the seq2seq model efficiently explore the vast vocabulary to learn new information and exploit the learnt information to predict correct words at each decoding step of the autoregressive question generation process.

- *Off-policy Critic training with an explicit division of high-reward and low-reward training samples:* We use off-policy training with experience replay for training the Critic to prevent the Critic from overfitting the sequence of on-policy training data. We use a prioritized replay buffer to store training samples for the off-policy Critic training, which helps break the correlation in the training data and reuse the samples for training. As an added improvement, we divide the training samples into high-reward and low-reward experiences based on the true lookahead return. This division in the samples arguably helps improve the surface-level similarity between the generated questions and the ground truth, evident from increased corpus-level scores.

## 2 Methodology

### 2.1 Automatic QUestion Generation

We define the problem of Automatic Question Generation (AQG) in the supervised learning setup as the problem of finding the parameter $\theta^*$, which maximizes the likelihood of observing the ground-truth sequence $Y$ for a given input $X = f(P, A)$ comprised of a passage $P$ and an answer $A$. Formally, we can state it as follows:

$$\theta^* = \arg\max_{\theta} \ln \pi_\theta(Y|X) \tag{1}$$

Here, $\pi_\theta(Y|X)$ is the probability of observing the ground-truth sequence $Y$ given the passage-answer text $X$.

#### 2.1.1 AQG as a Markov Decision Process

We often formulate sequential decision-making-based reinforcement learning problems as Markov Decision Processes (MDPs). Formally, an MDP is a tuple $(\mathcal{S}, \mathcal{A}, \mathcal{P}, \mathcal{R}, \gamma)$ where:

- $\mathcal{S}$ is the finite set of states,

- $\mathcal{A}$ is the finite set of actions,

- $\mathcal{P} : \mathcal{S} \times \mathcal{A} \times \mathcal{S} \to \mathcal{R}$ is the state transition function,

- $\mathcal{R} : \mathcal{S} \times \mathcal{A} \to \mathcal{R}$ is the reward function, and

- $\gamma \in [0, 1]$ is the discount factor.

In the MDP setting, we define the agent's goal as taking a sequence of actions over the finite horizon of discrete time steps to optimize for a task-specific goal. Specifically, at any given time step $t$, the agent observes the state $S_t$ of the environment, takes action $A_t$ and receives a reward $R_t$ for taking action the from that state. This process repeats for $T$ steps, and the agent's goal is to maximize the discounted sum of rewards received over the horizon $T$ and learn the *optimal policy* $\pi^*$ in the process.

To leverage Actor-Critic methods from the reinforcement learning literature for AQG, we re-formulate the AQG task as a sequential MDP, where an agent interacts with an environment over discrete time steps in a given episode, executing a sequence of actions to learn an optimal policy $\pi^*$, which maximizes some linguistic aspect-specific reward corresponding to the question generated autoregressively by the seq2seq model in that episode. In this setting:

- Each decoding step is a unique time step of an episode of the AQG MDP.

- The input passage-answer text $X$ and question sub-sequence $\hat{Y}_{\{1,\ldots,t-1\}}$ generated till the start of the time step (*i.e.,* decoding step) $t$ represents the state of the environment at time step $t$.

- The Actor and Critic models comprise the agent. Specifically, the Actor is responsible for learning the policy $\pi_\theta$ and taking action. The Critic is responsible for learning the state value function $V_\psi$ and generating the advantage estimates to evaluate the Actor's decisions.

- The token $\hat{y}_t$ output by the Actor at every decoding step $t$ after observing the state is the action $a_t$ taken by the agent at that time step. The decoding stops once the Actor generates the $< EOS >$ token or if it has already output the maximum permissible number of tokens.

- The sequence of actions the agent executes in this scenario is the sequence of tokens generated by the seq2seq model during decoding.

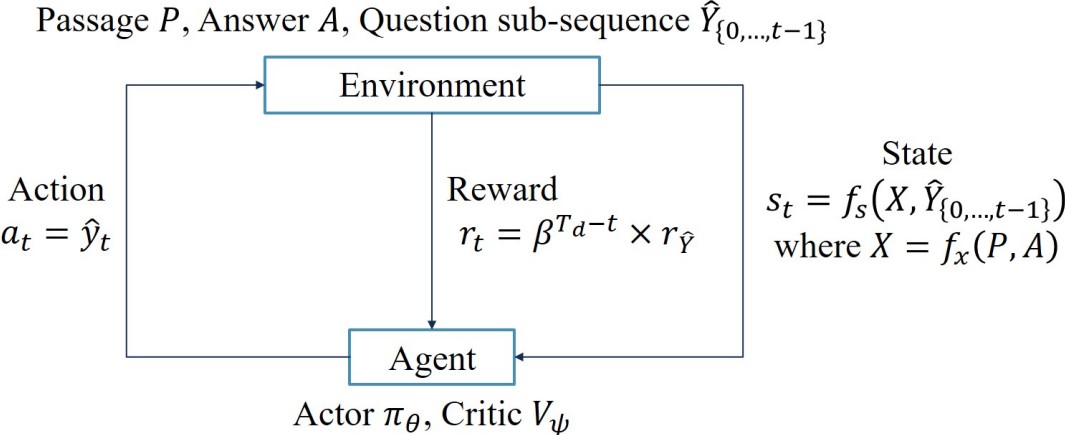

Figure 1: MDP formulation of the proposed Advantage Actor-Critic training framework for AQG.

- The policy tries to maximize the BLEU score, a surface-level metric that rates questions with more similarity with the ground-truth target questions higher than those with less similar ones. However, instead of directly maximizing the linguistic aspect-specific reward, the learned policy in our proposed Advantage Actor-Critic training framework tries to maximize the advantage estimates returned by the Critic at every decoding step, a strategy which helps address the global reward problem commonly observed in the case of Monte Carlo return-based policy gradient methods like REINFORCE with baseline.

Figure 1 illustrates the MDP used in our Advantage Actor-Critic framework.

## 2.2 Advantage Actor-Critic Agent

We follow an architecture-agnostic approach concerning the architecture of the seq2seq model in our proposed framework. The primary motivation behind the architecture-agnostic approach is to ensure that the framework is robust and can easily translate to different seq2seq models, which might vary from each other depending on their exact architectures. As a result, we can use our framework to train any typical seq2seq model comprising encoding and decoding components. In general, we use a shared network architecture for our Advantage Actor-Critic agent, where we share the lower layers of the Actor and Critic models to learn a common representation of the state space. This sharing of parameters between the Actor and Critic helps reduce the number of learnable parameters, thereby improving the sample efficiency during training. The foundational network used in the current work is a typical seq2seq transformer model (Vaswani et al., 2017) comprised of self-attention-based encoding and decoding components. The language modelling head of the seq2seq model is our Actor, which generates the probability distribution over the set of all tokens in our vocabulary and learns the policy. The Critic is a simple feedforward neural network which generates the advantage estimates which provide implicit information about the question-generation process even before completion. We follow a similar input pipeline like the *BERT-HLSQG* model (Chan & Fan, 2019) to feed the passage-answer text input to the transformer-based seq2seq model. In this approach, we add a special $< hl >$ token at the beginning and end of the answer span to indicate the answer sequence in the input passage. The $< hl >$ token removes any ambiguity arising from the presence of the answer sequence in multiple places in the input passage, helping the seq2seq focus on the relevant section of the passage while generating the question. At every decoding step, the state representation module part of the seq2seq model generates a contextualized vector representation $s_t = f_s(X, \hat{Y}_{\{0,\dots,t-1\}})$, which is a function of the input passage-answer text $X$ and the question prefix $\hat{Y}_{\{0,\dots,t-1\}}$ generated till the start of that decoding step $t$. In fact, $s_t$ is the state the Actor and Critic observe at that time step $t$. The Actor, which is the language modelling head of the seq2seq model, outputs a token $\hat{y}_t$ based on the observed state $s_t$, and the Critic estimates the state-value

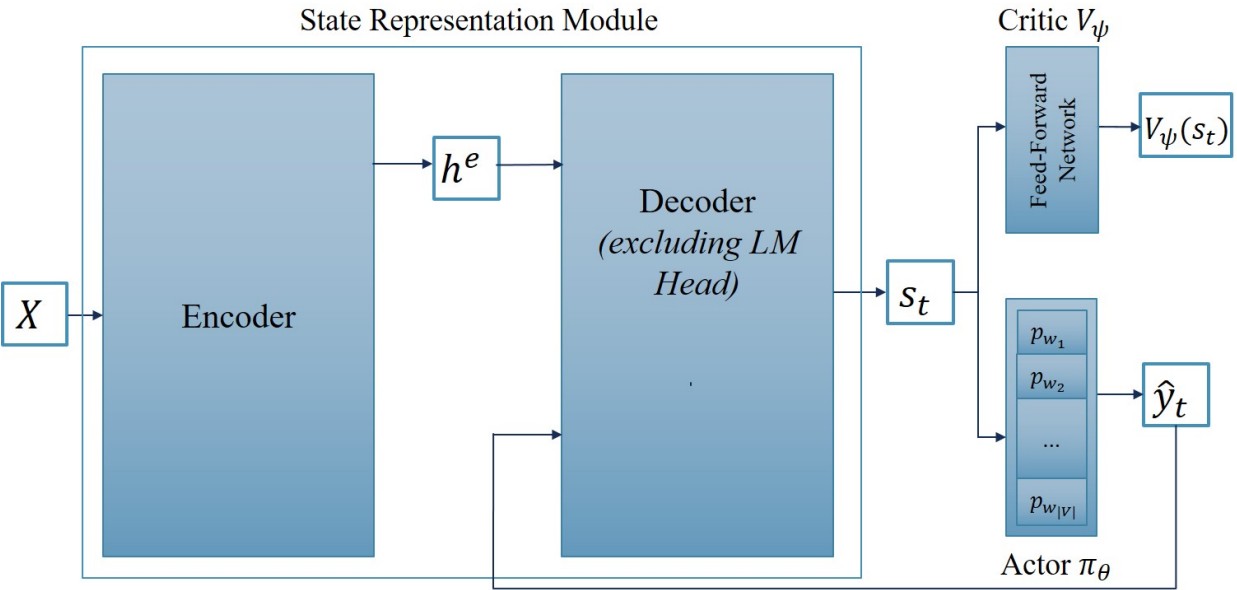

Figure 2: Overview of the Advantage Actor-Critic agent comprising of the Actor and the Critic.

Figure 3: In our proposed Advantage Actor-Critic training framework, the language modelling head of the sequence-to-sequence model is the Actor, which is responsible for learning the policy $\pi_\theta$.

$V_\psi(s_t)$ for the observed state $s_t$. Figure 2 gives an overview of the Advantage Actor-Critic agent comprising the Actor and the Critic.

## 2.3 Reward Function

The reward function is vital to the training process as it incentivizes the model to generate questions with specific characteristics. In our proposed Advantage Actor-Critic training framework, we define a surface-level similarity-based reward function that ensures that the Actor's questions have significant structural similarity with the ground-truth questions, improving overall fluency. Nema & Khapra (2018) define fluency in terms

Figure 4: In our Advantage-Actor-Critic training framework, we use a simple feed-forward neural network as the Critic, which is responsible for learning the state-value function $V_\psi$. We then use the state-value estimates from the Critic to compute the advantage function $A_\psi$.

of BLEU scores. We use a similar definition of fluency. However, the BLEU metric is a corpus-level similarity metric unsuitable for generating sentence-level scores. Therefore, we use smoothing (Chen & Cherry, 2014) on top of the vanilla BLEU score computation for our Actor-Critic training.

Although the reward function awards the agent for generating questions with more significant surface-level similarity, simply using the sentence-level score *i.e.,* the global reward computed only at the end of the question generation process) can be inefficient. The intermediate decoding steps get zero rewards in such a setting, leading to a sparse training target for the Critic. Therefore, as an alternative, we define a per-step reward function that provides dense targets for training the Critic. Specifically, we use a lookahead rewards-based scoring mechanism which generates a scalar lookahead reward upon each addition of a new token. We take the help of the sentence-level fluency score and use it to compute the per-step lookahead reward at each decoding step $t$ as follows:

$$r_t = \beta^{T_d - t} \times r_{\hat{Y}} \tag{2}$$

Here, $T_d$ is the total number of decoding steps, $r_{\hat{Y}}$ is the sentence-level fluency score corresponding to the complete question, and $\beta \in [0, 1]$ is a hyper-parameter that controls the confidence of the Actor about the future sentence-level score at intermediate time step $t$. $\beta = 1$ indicates that the Actor is fully confident about the sentence-level reward even at the intermediate step of the question generation process. In comparison, $\beta < 1$ indicates that the Actor is less confident about the global reward at the beginning, and the confidence grows as the Actor adds more and more tokens to the question sequence at each decoding step. The primary idea behind using the above approach for the lookahead reward calculation is that since the tokens accumulate over multiple decoding steps to generate the complete question, the sentence-level scores provide latent information about the intermediate sequences composing the final question. Low scores indicate the accumulation of inferior tokens and vice versa. Therefore, we can use the sentence-level score as a surrogate for the per-step reward and provide dense targets for training the Critic, which helps compute proper advantage estimates.

## 2.4 Training using A2C with GAE

We use the Advantage Actor-Critic (A2C) algorithm with Generalized Advantage Estimation (GAE) (Schulman et al., 2016) to train the Actor and Critic models.

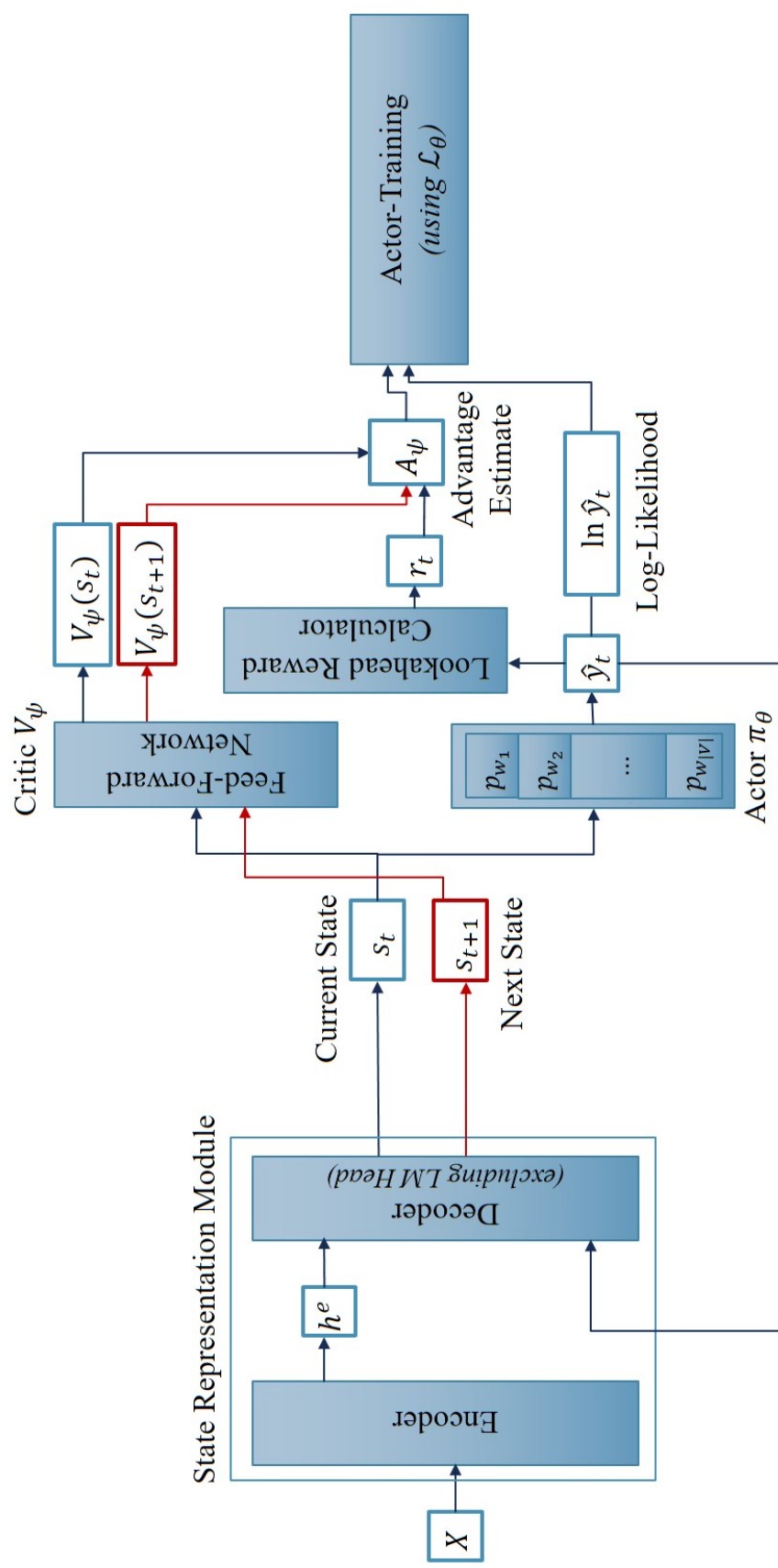

Figure 5: Overview of the Actor training step of the Advantage Actor-Critic training framework. The Actor outputs action $\hat{y}_t$ from state $s_t$ based on which the Critic returns the state value estimates $V_\psi(s_t)$ and $(V_\psi(s_{t+1})$ for the current and the next state. We then use these estimates for computing the 1-step advantage estimate $A_{\psi(s_t)}$ (assuming GAE parameter $\lambda = 0$ and update the Actor parameter $\theta$ using the loss $\mathcal{L}_\theta$, based on the log-likelihood $\ln y_t$ and the advantage estimate $A_{\psi(s_t)}$.

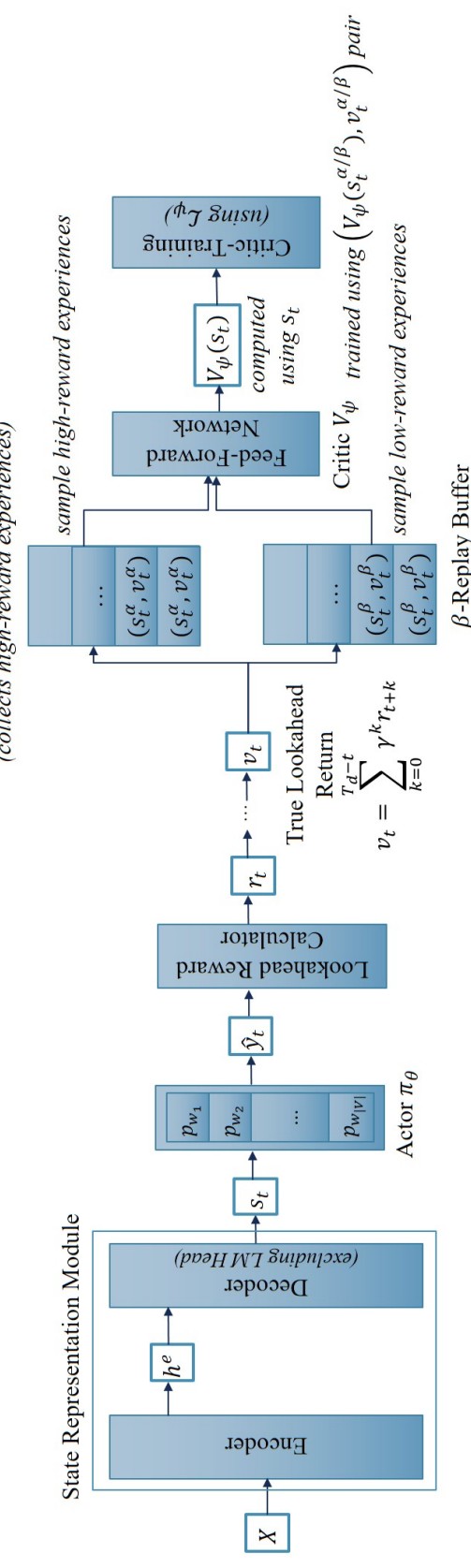

Figure 6: Overview of the Critic training step of the Advantage Actor-Critic training framework. We store the state $s_t$ and the corresponding true lookahead return $v_t$ as a pair, called an experience $e_t = (s_t, v_t)$, in separate replay buffers, $\alpha$-replay buffer and $\beta$-replay buffer, depending on whether the experiences are high-reward or low-reward experiences. When the replay buffers are sufficiently full, we sample an equal-sized batch of experiences from both replay buffers and compute the state value estimate $V_\psi(s_t)$ for each training sample. We then update the Critic parameter $\psi$ using the MSE-based loss $\mathcal{L}_\psi$.

We follow an on-policy approach for training the Actor, where the Actor samples a sequence of tokens using its decoder's probability distribution and uses the normalized advantage estimates from the Critic to update the Actor parameter $\theta$ by maximizing the following policy loss:

$$\mathcal{L}_\theta^{AC} = \frac{1}{N_q} \sum_{i=1}^{N_q} \sum_{t=1}^{T_d} \log \pi_\theta(\hat{y}_{i,t}|s_{i,t}) \hat{A}_\psi^{GAE}(s_{i,t}, \hat{y}_{i,t}) \tag{3}$$

Here $s_{i,t}$ is the state at the $t^{th}$ decoding step for the $i^{th}$ sample, $\hat{y}_{i,t}$ is the token predicted by the Actor from the state $s_{i,t}$, and $\hat{A}_\psi^{GAE}(s_{i,t}, \hat{y}_{i,t})$ is the normalized advantage score for the state-action pair $(\mathbf{s}_{i,t}, \hat{y}_{i,t})$ estimated using the GAE trick, which is a temporal-difference (TD) method.

For training the Critic, we follow an off-policy approach, where we collect experiences of the form $e_t = (s_t, v_t)$ in a replay buffer $\mathcal{D}$ for training the Critic. Here, $s_t$ is the observed state, and $v_t$ is the target state-value based on the true lookahead return. Once $\mathcal{D}$ contains a sufficiently large number of experiences, we sample a mini-batch of size $N_q$ and use the sampled experiences to update the Critic parameter $\psi$ by minimizing the following Mean Squared Error (MSE)-based value loss:

$$\mathcal{L}_\psi^{AC} = \frac{1}{N_q} \sum_{i=1}^{N_q} ||v_i - V_\psi(s_i)||^2 \tag{4}$$

Here, $V_\psi(s_i)$ is the state-value estimated by the Critic network for the state $s_i$, and $v_i$ is target state-value used for computing the value loss. As mentioned earlier, we use the true lookahead return at time $t$ to compute the target state-value $v_i$ as follows:

$$v_i = \sum_{k=t}^{T_d} r_{i,k} \tag{5}$$

Since we use experiences collected from the Actor's prediction to compute $\mathcal{L}_\theta$, the training becomes completely auto-regressive and independent of the ground-truth tokens, removing exposure bias. In addition, the advantage estimates from the Critic, computed using the true lookahead return, provide evaluation metric-based guidance while training the Actor, addressing the evaluation measure mismatch problem. The Critic also uses the TD method-based GAE model to generate the generalized advantage estimate for every token output by the Actor, which helps address issues associated with global rewards like sparse training signals and high variance.

## 2.5 Intractable Action Space Bottleneck

The vast vocabulary from which the Actor outputs tokens poses a significant challenge when extending reinforcement learning methods like our proposed Advantage Actor-Critic training framework for sequence generation tasks like AQGs. The token sequence is the agent's actions executed over an entire decoding episode. Given the discrete and vast nature of the action space, the agent's ability to reason in such an environment becomes crucial to solving such sequence generation problems efficiently. However, the intractable nature of this vast action space often necessitates excessive training for the agent to even converge to a remotely favourable policy. In the worst case, it can even result in the agent failing to find an appropriate policy. Therefore, the *intractable action space problem* is a severe bottleneck that makes reinforcement learning-based training difficult.

### 2.5.1 Proposed Workaround

We can easily observe that only a small subset of tokens is valid at any step of the question-generation process. Therefore, the agent needs to only search a subset of the vocabulary for valid tokens and output tokens from that relevant subset at each decoding step. Such an approach allows the agent to explore the action space efficiently to find the optimal action. Therefore, we propose a workaround based on the said approach, which is composed of the following two tasks:

- *Multi-step training to learning about the relevant subset of the vocabulary:* In our proposed framework, we learn about the relevant subset of valid tokens by dividing the overall training into multiple steps and including a supervised pre-training step in our multi-step training process. The primary motivation is that the supervised pre-training step helps recognize the relevant subset, allowing for a more focused exploration of the vocabulary space. An added advantage of this strategy is the apparent speed in training due to the prevention of the mindless generation of garbage sequences in the initial training phase.

- *Probability-based action pruning for efficient exploration by exploiting the learnt information:* Once the supervised pre-training is over and the agent learns crucial information about the vocabulary, we exploit the learnt information in a joint training step by proposing an alternative exploration strategy instead of the usual softmax-based sampling. Specifically, we propose using probability-based action pruning strategies like *Top-K* and *Top-p* sampling, which allows the agent to explore the vocabulary in a restricted but efficient manner based on the probability distribution over the tokens learnt during the supervised pre-training step.

### 2.5.2 Multi-step Training Process

We discussed that although the vocabulary is vast, only a small subset of tokens is valid at every decoding step. Starting the training with an untrained Actor, which randomly samples tokens from the entire vocabulary, can be impractical. Instead, we can provide some external information about the vocabulary to the Actor, which helps it only spend a small amount of time initially exploring the vast vocabulary to learn very little information. Therefore, we follow a multi-step training process where we divide the entire procedure into the following three steps:

1. *Supervised pre-training of the Actor:* We pre-train the Actor initially in a supervised learning setup using cross-entropy loss, where the Actor uses the ground-truth output tokens to update its parameters. This step helps expedite the training and prevent the Actor from mindlessly generating garbage sequences in the initial training phase. This step updates the entire seq2seq model parameters responsible for state representation and action selection.

2. *Critic pre-training using Actor's predictions:* We generate tokens using the Actor's predictions and pre-train the Critic to minimize the value loss $\mathcal{L}_\psi$. We only update the parameters of the Critic in this step keeping the Actor parameters fixed.

3. *Actor-Critic joint training using A2C with GAE:* We finally do the Actor-Critic joint training, where we simultaneously update the Actor and Critic parameters. We keep all the parameters of the seq2seq model fixed other than those of the language modelling head to feed consistent state representations to the Actor and the Critic. The A2C with GAE-based joint training leads to a hard transition from ground-truth token-based training to absolute training with the Actor's predictions. It helps remove exposure bias and exploit the valuable information about the vocabulary learnt from the pre-training stage.

### 2.5.3 Probability-based Action Pruning

After completing the supervised pre-training step, the agent has acquired crucial information about the subset of valid tokens in the vocabulary. Softmax-based token sampling fails to take full advantage of this information. It can also generate incoherent and gibberish sequences (Schulman et al., 2016), which are not desirable. Other alternative text generation strategies like greedy search and beam search are also not suitable because these do not generate diverse training sequences for the Advantage Actor-Critic agent.

Top-$K$ and Top-$p$ sampling are simple and effective solutions that can exploit valuable vocabulary information by pruning the more extensive vocabulary and generating diverse coherent training samples from a smaller subset of valid tokens for the reinforcement learning agent. People already use these sampling strategies during inferencing to generate diverse sequences in open-ended text generation tasks. We propose using such strategies in the training phase to generate valid desired training samples to help train our agent to generate appropriate questions. We briefly describe these sampling strategies as follows:

- *Top-K sampling:* (Fan et al., 2018) introduced Top-$K$ sampling, which samples tokens from a subset of $K$ most likely tokens at every decoding step. Specifically, at every decoding step, we filter the $K$ most likely tokens, redistribute the probability mass among these $K$ tokens, and allow the decoder to sample a token from this smaller set. Since the decoder can only choose from this smaller set, we restrict the agent's exploration to the $K$ most likely actions. This strategy is very effective because it allows the agent to save time by exploring only the relevant parts of the vocabulary by focusing its attention on the smaller set of valid tokens.

  However, the Top-$K$ sampling filters a fixed number of tokens at every decoding step and cannot dynamically adapt the number of tokens according to the input. As a result, in scenarios where the probability distribution over the vocabulary is relatively flat, Top-$K$ sampling might miss out on reasonably good tokens. Similarly, while dealing with very sharp distributions, Top-$K$ sampling might redistribute the probability mass over completely irrelevant tokens, which unfortunately got filtered because of the fixed value of $K$.

- *Top-p Sampling:* (Holtzman et al., 2020) introduced Top-$p$ sampling, which can dynamically adapt the number of tokens to sample from at every decoding step instead of solely sampling from a fixed set of $K$ tokens. Specifically, Top-$p$ sampling filters the smallest feasible set of tokens whose cumulative probability surpasses a given threshold $p$, redistributes the probability mass among the subset and lets the decoder sample a token from this dynamically selected subset. Therefore, this technique efficiently increases or decreases the subset size dynamically according to the input of the decoding step and allows the agent to explore the vocabulary space.

### 2.6 Off-Policy Critic Training with Experience Replay

We use off-policy training with experience replay for training the Critic, where we store the state $s_t$ and the corresponding target state-value $v_t$ as a training sample pair $e_t = (s_t, v_t)$, called an experience, in a replay buffer. When the replay buffer has sufficient experiences, we sample a mini-batch of experience from the replay buffer, compute the state value estimate $V_\psi(s_t)$ for each training sample., and update the Critic parameter $\psi$ using the MSE-based loss $\mathcal{L}_\psi$. The primary motivation for using this off-policy approach for training the Critic is to prevent the Critic from overfitting the sequence of on-policy training data. Since subsequent states differ by a single token only, the sequence of training samples collected from the current decoding episode of the Actor is highly correlated. Therefore, training on a random set of experiences sampled from the replay buffer breaks this correlation. Another advantage is that it allows us to reuse the experiences to train the Critic. We use the prioritized experience replay (Schaul et al., 2016) strategy to improve the sampling from the replay buffer.

As an added improvement, we divide the training samples into high-reward and low-reward experiences based on the true lookahead return. We ensure we have an equal proportion of experiences from each type in the mini-batch for training the Critic. The primary reason for using such an approach is the observation that most of the sequences generated during training get low returns. Such states are not informative and impart less information to improve the question-generation process. In contrast, states with high returns are valuable samples which capture the sequence-level relation well and contain information that can help the Critic provide proper signals to the Actor and help improve the question-generation process. To implement this approach, we use two replay buffers, the $\alpha$-replay buffer and the $\beta$-replay buffer, separately storing the high-reward and low-reward experiences. When updating the Critic parameter $\theta$, we sample an equal number of samples from each replay buffer.

## 3 Experimental setup

### 3.1 Dataset

Ushio et al. (2022) introduced QG-Bench, a collection of several multilingual and multidomain Question-Answering (QA) datasets unified into a single benchmark for AQG research. Our experiments use the SQuAD, SQuADShifts, and SubjQA datasets from this benchmark:

Table 2: Statistics of SQuAD, SQuADShifts, and SubjQA datasets as reported in QG-Bench.

| Dataset | No. of Samples | | |
| --- | --- | --- | --- |
| | Training | Validation | Test |
| **SubjQA** | | | |
| **Books** | 637 | 92 | 191 |
| **Electronics** | 697 | 99 | 238 |
| **Grocery** | 687 | 101 | 379 |
| **Movies** | 724 | 101 | 154 |
| **Restaurants** | 823 | 129 | 136 |
| **Tripadvisor** | 875 | 143 | 397 |
| **SQuADShifts** | | | |
| **Amazon** | 3295 | 1648 | 4942 |
| **Wiki** | 2646 | 1323 | 3969 |
| **News** | 3355 | 1678 | 5032 |
| **Reddit** | 3268 | 1634 | 4901 |
| **SQuAD** | | | |
| **Paragraph-Level** | 75722 | 10570 | 11877 |
| **Sentence-Level** | 75722 | 10570 | 11877 |

- *SQuAD (Rajpurkar et al., 2016):* The Stanford Question Answering Dataset is a popular dataset used for AQG research. It contains more than 100K (question, answer) pairs collected by crowd-sourcing over a set of Wikipedia articles. The answers to the questions are also present in the articles as a sequence of tokens. QG-Bench also includes this dataset in its collection and uses the same data split as in Du et al. (2017).

- *SQuADShifts (Miller et al., 2020):* SQuADShifts is one of the two domain-specific datasets in QG-Bench. It contains questions similar to the SQuAD dataset but from four domains- Amazon, Wikipedia, News and Reddit.

- *SubjQA (Bjerva et al., 2020):* SubjQA is another domain-specific dataset in QG-Bench. This dataset contains subjective question-answer pairs collected from six domains- Books, Electronics, Grocery, Movies, Restaurants, and Tripadvisor.

We report the statistics of these datasets in Table 2.

### 3.2 Implementation Details

Our experiments use pre-trained transformer models from Ushio et al. (2022). These models are already pre-trained in the supervised learning setup using the teacher-forcing algorithm and provide a convenient way to proceed with the multi-step training process used in our proposed Advantage Actor-Critic training framework. Since we follow an architecture-agnostic approach in our work, we primarily use the pre-trained $T5_{SMALL}$ model in our experiments and limit our focus to testing the ability of the Advantage Actor-Critic training in improving the performance of the same model for AQG. The $T5_{SMALL}$ model has $\sim 60$ million parameters and is far smaller than the other state-of-the-art transformer models used for AQG (Dong et al. (2019), Qi et al. (2020), Ushio et al. (2022)). An added advantage of using the $T5_{small}$ is using a subword-level vocabulary consisting of 32000 unique tokens, which is smaller than a typical word-level vocabulary. This subword-level vocabulary also removes the need for the maxout pointer network and helps simplify the task significantly. We also use a simplified Critic in the Advantage Actor-Critic framework, which generates a scalar state-value estimate at every time step. We use a 3-layer, fully-connected feedforward neural network (FNN) for the Critic, where the two hidden layers have 256 and 128 units, respectively. The output layer generates state-value estimates for the state representations obtained from the state-representation module. The size of the state representations in the current work is 512, but it can be increased or decreased depending on the specific architectural requirements.

The details of the remaining training steps of our multi-step training process are as follows:

- *Critic pre-training using Actor's predictions:* We pre-train the Critic for a maximum of 50 epochs using the AdamW optimizer (Loshchilov & Hutter, 2019). We initialize the learning rate to 0.01 and keep it constant for the entire duration of the Critic pre-training. We experiment with Top-$K$ and Top-$p$ sampling for generating questions. We collect training samples in separate prioritized replay buffers, $\alpha$-replay buffer and $\beta$-replay buffer, where we explicitly separate the good and bad experiences based on a threshold lookahead return of 0.75. We sample mini-batches containing 128 experiences from both replay buffers to ensure that each batch comprises an equal number of high-reward and low-reward experiences for training the Critic. We keep the Actor's parameter fixed during this step.

- *Actor-Critic joint training using A2C with GAE:* We use the AdamW optimizer (Loshchilov & Hutter, 2019) with a linear decaying schedule for both the Actor and the Critic models and initialize the learning rate to either 0.00005 (for SQuAD-related experiments) or 0.0005 (for SQuADShifts and SubjQA-related experiments) for the Actor optimizer and 0.001 for the Critic optimizer. Like the Critic pre-training step, we also experiment with Top-$K$ and Top-$p$ sampling for generating questions and employ an off-policy approach with experience replay for training the Critic. We set the lookahead confidence parameter to 1.0, indicating the agent is fully confident about the sentence-level reward even at the intermediate steps. Unlike the Critic pre-training step, we update the Actor model in an on-policy manner. In our experiments, we set the GAE parameter $\lambda = 0$ and use 1-step advantage estimates to measure any action's advantage. Based on hyperparameter tuning, we limit the training to a maximum of 5 epochs (for SQuAD and SQuADShifts-related experiments) and 30 epochs (for SubjQA-related experiments).

After training, we select the model which obtains the highest corpus-level BLEU-4 score using beam search decoding on the validation data and report results obtained by the model on the test data. We use a beam size of 50 in our experiments. We also use an n-gram penalty mechanism (Paulus et al., 2018), where we look at the possible sequence of the following words that might result in bi-gram repetitions and manually set their probabilities to zero. This strategy ensures that no bi-gram appears twice while generating the question and removes the repetitive phrase problem commonly observed in sequence generation tasks.

### 3.3 Evaluation

We evaluate the models' performance using our proposed Advantage Actor-Critic training framework on the BLEU-4 (Papineni et al., 2002) evaluation metric.

## 4 Results and Discussions

In this section, we study the efficacy of our proposed Advantage Actor-Critic training framework and analyze the different design choices proposed.

### 4.1 Analysis of Lookahead Reward Function

Here, we analyze the lookahead reward function and its effect on the overall Advantage Actor-Critic training by comparing it with the global rewards strategy, where sparse global rewards drive the training. All decoding steps except the final one get a zero reward in the global rewards strategy, an approach which is different from the per-step lookahead rewards-driven training. Table 3 shows that we obtain higher corpus-level BLEU-4 scores using the lookahead reward function on all the datasets except the Restaurant domain of SubjQA dataset, where both reward functions result in a score of zero. We observe that lookahead rewards-driven training gets a higher corpus-level BLEU-4 score by 0-3% across all the datasets, apart from the Books domain of the SubjQA, where the improvement is $\sim 17\%$. This consistent trend of higher corpus-level BLEU-4 scores across the different datasets provides empirical proof regarding the benefit of lookahead rewards over the vanilla global reward mechanism.

Table 3: Corpus-level BLEU-4 obtained using the global and the lookahead reward functions for the Advantage-Actor-Critic training.

| Dataset | Reward Function | |
|---|---|---|
| | Global | Lookahead |
| **SubjQA** | | |
| **Books** | 7.11 | 8.31 |
| **Electronics** | 25.82 | 26.04 |
| **Grocery** | 13.82 | 14.66 |
| **Movies** | 13.84 | 13.84 |
| **Restaurants** | 0.00 | 0.00 |
| **Tripadvisor** | 18.07 | 19.69 |
| **SQuADShifts** | | |
| **Amazon** | 15.96 | 16.23 |
| **Wiki** | 23.47 | 23.54 |
| **News** | 21.07 | 21.37 |
| **Reddit** | 15.00 | 15.39 |
| **SQuAD** | | |
| **Paragraph-Level** | 18.58 | 18.86 |
| **Sentence-Level** | 17.47 | 17.92 |

Table 4: Comparison of untrained Actor-based joint training and proposed multi-step training regarding corpus-level BLEU-4 scores.

| Dataset | Training | | |
|---|---|---|---|
| | **Untrained Actor** | **Supervised Pre-training** | **Actor-Critic Joint Training** |
| **SubjQA** | | | |
| **Books** | 0.00 | 7.01 | 8.31 |
| **Electronics** | 0.00 | 25.82 | 26.04 |
| **Grocery** | 0.00 | 13.81 | 14.66 |
| **Movies** | 0.00 | 13.84 | 13.84 |
| **Restaurants** | 0.00 | 0.00 | 0.00 |
| **Tripadvisor** | 0.00 | 17.88 | 19.69 |
| **SQuADShifts** | | | |
| **Amazon** | 0.37 | 16.02 | 16.23 |
| **Wiki** | 1.87 | 23.49 | 23.54 |
| **News** | 1.23 | 21.14 | 21.37 |
| **Reddit** | 0.57 | 15.20 | 15.39 |
| **SQuAD** | | | |
| **Paragraph-Level** | 1.77 | 18.65 | 18.86 |
| **Sentence-Level** | 4.50 | 17.55 | 17.92 |

## 4.2 Analysis of Multi-Step Training Procedure

Here, we analyze the multi-step training procedure. We compare the results of untrained Actor-based joint training and the multi-step training in Table 4. As expected, the training either fails or learns very slowly when we start with an untrained Actor. The untrained Actor takes a long time to generate meaningful tokens from the vast vocabulary and wastes a significant duration generating garbage sequences. The pre-training step accelerates the training by guiding the Actor to focus on the relevant part of the vocabulary. We obtain a high corpus-level BLEU-4 score by fine-tuning the pre-trained model in the following joint-training step, which provides BLEU-based optimization and helps improve the BLEU-4 scores further.

Table 5: Corpus-level BLEU-4 score obtained using Top-$K$, Top-$p$ and softmax sampling strategy. For Top-$K$ and Top-$p$ sampling, we report the best BLEU-4 score obtained by sampling tokens from a smaller subset of the vocabulary; for softmax sampling, we report BLEU-4 scores obtained by sampling tokens from the entire vocabulary.

| Dataset | Sampling | | |
|---|---|---|---|
| | Top-$K$ | Top-$p$ | Softmax |
| **SubjQA** | | | |
| Books | 8.31 | 8.04 | 8.11 |
| Electronics | 26.04 | 25.98 | 25.87 |
| Grocery | 14.42 | 14.66 | 14.48 |
| Movies | 13.83 | 13.83 | 13.84 |
| Restaurants | 0.00 | 0.00 | 0.00 |
| Tripadvisor | 19.45 | 19.69 | 19.35 |
| **SQuADShifts** | | | |
| Amazon | 16.23 | 16.20 | 16.09 |
| Wiki | 23.54 | 23.43 | 23.42 |
| News | 21.37 | 21.36 | 21.24 |
| Reddit | 15.28 | 15.36 | 15.39 |
| **SQuAD** | | | |
| Paragraph-Level | 18.85 | 18.86 | 18.86 |
| Sentence-Level | 17.92 | 17.92 | 17.81 |

Table 6: Corpus-level BLEU-4 score obtained for different values of $K$ of Top-$K$ sampling.

| Dataset | $K$ | | | All Tokens |
|---|---|---|---|---|
| | 10 | 100 | 1000 | |
| **SubjQA** | | | | |
| Books | 7.06 | 8.31 | 7.36 | 8.11 |
| Electronics | 25.77 | 25.56 | 26.04 | 25.87 |
| Grocery | 14.15 | 14.42 | 14.30 | 14.48 |
| Movies | 13.82 | 13.82 | 13.83 | 13.84 |
| Restaurants | 0.00 | 0.00 | 0.00 | 0.00 |
| Tripadvisor | 18.84 | 19.45 | 19.32 | 19.35 |
| **SQuADShifts** | | | | |
| Amazon | 16.14 | 16.06 | 16.23 | 16.09 |
| Wiki | 23.39 | 23.54 | 23.38 | 23.42 |
| News | 21.37 | 21.18 | 21.25 | 21.24 |
| Reddit | 15.19 | 15.24 | 15.28 | 15.39 |
| **SQuAD** | | | | |
| Paragraph-Level | 18.84 | 18.85 | 18.79 | 18.86 |
| Sentence-Level | 17.84 | 17.92 | 17.88 | 17.81 |

## 4.3 Analysis of Probability-based Action Pruning

Here, we study the different probability-based action pruning strategies and analyze how Top-$K$ and Top-$p$ sampling help our Actor-Critic agent generate good-quality questions by effectively exploring the vocabulary space in a restricted but efficient manner. We report the corpus-level BLEU-4 score obtained using different sampling strategies in Table 5. For Top-$K$ and Top-$p$ sampling, we report the best BLEU-4 score obtained by sampling tokens from a smaller subset of the vocabulary. In contrast, we sample tokens from the entire vocabulary during softmax sampling. Additionally, we also report the corpus-level BLEU-4 score obtained for different values of $K$ and $p$ in Table 6 and Table 7 respectively. We observe no specific value of $K$ or $p$, consistently giving the best results across all datasets. We, however, observe that using a smaller subset of the vocabulary often gives higher or comparable scores compared to training with the whole vocabulary.

Table 7: Corpus-level BLEU-4 score obtained for different values of $p$ of Top-$p$ sampling.

| Dataset | $p$ | | | All Tokens |
|---|---|---|---|---|
| | **0.850** | **0.875** | **0.900** | |
| **SubjQA** | | | | |
| **Books** | 8.04 | 7.10 | 7.09 | 8.11 |
| **Electronics** | 25.98 | 25.91 | 25.75 | 25.87 |
| **Grocery** | 14.43 | 14.66 | 14.40 | 14.48 |
| **Movies** | 13.81 | 13.82 | 13.83 | 13.84 |
| **Restaurants** | 0.0 | 0.00 | 0.00 | 0.00 |
| **Tripadvisor** | 19.46 | 19.49 | 19.69 | 19.35 |
| **SQuADShifts** | | | | |
| **Amazon** | 16.20 | 16.17 | 16.14 | 16.09 |
| **Wiki** | 23.43 | 23.34 | 23.40 | 23.42 |
| **News** | 21.36 | 21.34 | 21.22 | 21.24 |
| **Reddit** | 15.36 | 15.34 | 15.28 | 15.39 |
| **SQuAD** | | | | |
| **Paragraph-Level** | 18.85 | 18.81 | 18.86 | 18.86 |
| **Sentence-Level** | 17.87 | 17.79 | 17.92 | 17.81 |

For example, we get a corpus-level BLEU-4 score of 8.31 on the Amazon domain using $K = 100$, while the same training using the complete vocabulary results in a lower score of 8.11.

However, we can see that pruning the vocabulary too much can be counter-productive, a trend confirmed by the lower corpus-level BLEU-4 score obtained when sampling with $K = 10$. Such a small value of $K$ significantly restricts exploration, as a result of which, the agent settles down on a sub-optimal question generation policy. Although Top-$p$ sampling should ideally result in better corpus-level BLEU-4 scores because of its ability to dynamically update the list of valid tokens at every decoding step, Table 5 shows that the trend is not consistent. One reason behind this might be the fact that even higher-order $p$-values (*e.g.,* 0.8) often include too few tokens to drive efficient exploration.

### 4.4 Analysis of Critic Training with $\alpha$- and $\beta$-Replay Buffers

Here, we analyze the effect of separate replay buffers, used for separately storing and sampling high-reward and low-reward experiences, on the overall training performance. We mentioned in Section 2.6 that we use this approach to explicitly ensure that the mini-batch for training the Critic has an equal number of high-reward and low-reward samples to help the Critic learn the state-values efficiently. Table 8 reports the corpus-level BLEU-4 score obtained for mini-batches having different fractions of high-reward experiences. We specifically test mini-batches containing 1%, 50% and 99% high-reward samples. Also, we use a vanilla softmax sampling strategy and consider the whole vocabulary for generating the question sequences for creating the Critic's training samples. We can see from Table 8 that the overall corpus-level BLEU-4 improves as the percentage of high-reward experiences in the mini-batch increases, evident from the consistent trend of the lowest corpus-level score obtained for mini-batches having only 1% of high-reward experiences. Another exciting trend is the additional improvement in corpus-level BLEU-4 when we favour high-reward samples in the mini-batches by increasing the percentage from 50% to 99%. We observe that the Actor model receives higher corpus-level scores across all but one dataset in such a scenario, pointing towards the fact that the high-reward samples capture sequence-level relations much better and pass on relevant information that can help the Critic provide proper signals to the Actor and help improve the question-generation process.

### 4.5 Advantage Actor-Critic Training vs Existing Training Frameworks

Here, we compare the Actor-Critic training with the widely used supervised training and PG-based SCST to study the advantages and disadvantages of our proposed approach.

Table 8: Corpus-level BLEU-4 score obtained for mini-batches having different fractions of high-reward experiences. We use the vanilla softmax sampling strategy of selecting tokens from the entire vocabulary for generating the Critic training samples.

| Dataset | Fraction of High-Reward Samples | | |
|---|---|---|---|
| | 0.01 | 0.50 | 0.99 |
| **SubjQA** | | | |
| **Books** | 7.79 | 8.11 | 8.52 |
| **Electronics** | 25.67 | 25.87 | 26.28 |
| **Grocery** | 14.13 | 14.48 | 15.78 |
| **Movies** | 13.81 | 13.84 | 13.91 |
| **Restaurants** | 0.00 | 0.00 | 0.00 |
| **Tripadvisor** | 17.39 | 19.35 | 20.60 |
| **SQuADShifts** | | | |
| **Amazon** | 16.09 | 16.09 | 16.15 |
| **Wiki** | 23.34 | 23.42 | 23.48 |
| **News** | 21.20 | 21.24 | 21.38 |
| **Reddit** | 15.39 | 15.39 | 15.40 |
| **SQuAD** | | | |
| **Paragraph-Level** | 18.84 | 18.86 | 18.90 |
| **Sentence-Level** | 17.76 | 17.81 | 17.86 |

Table 9: Comparison of different training frameworks based on corpus-level BLEU-4 scores.

| Dataset | Training Framework | | |
|---|---|---|---|
| | Supervised | SCST | Advantage Actor-Critic |
| **SubjQA** | | | |
| **Books** | 7.01 | 7.10 | 8.31 |
| **Electronics** | 25.82 | 25.77 | 26.04 |
| **Grocery** | 13.81 | 11.76 | 14.66 |
| **Movies** | 13.84 | 13.84 | 13.84 |
| **Restaurants** | 0.00 | 0.00 | 0.00 |
| **Tripadvisor** | 17.88 | 19.04 | 19.69 |
| **SQuADShifts** | | | |
| **Amazon** | 16.02 | 16.13 | 16.23 |
| **Wiki** | 23.49 | 23.49 | 23.54 |
| **News** | 21.14 | 21.14 | 21.37 |
| **Reddit** | 15.20 | 15.29 | 15.39 |
| **SQuAD** | | | |
| **Paragraph-Level** | 18.65 | 18.65 | 18.86 |
| **Sentence-Level** | 17.55 | 17.61 | 17.92 |

Table 10: Examples from SQuAD dataset where Advantage Actor-Critic training generates better questions than supervised training in terms of surface-level similarity with the ground truth.

| Example 1 | |
|---|---|
| **Paragraph:** | *… In the ceremony held at the Hong Kong Cultural Centre in Tsim Sha Tsui, Chief Executive Donald Tsang handed the torch to the first torchbearer, Olympic medalist* *Lee Lai Shan. …* |
| **Target:** | Who was the first torchbearer in Hong Kong? |
| **Supervised:** | Who was the first torchbearer? |
| **Advantage Actor-Critic:** | Who was the first torchbearer in Hong Kong? |
| **Example 2** | |
| **Paragraph:** | *Besides parents, Liu Shaokun,* *a Sichuan school teacher**, was detained on June 25, 2008 for "disseminating rumors and destroying social order" about the Sichuan earthquake. …* |
| **Target:** | What was Liu Shaokun's profession? |
| **Supervised:** | Who was Liu Shaokun? |
| **Advantage Actor-Critic:** | What was Liu Shaokun's profession? |
| **Example 3** | |
| **Paragraph:** | *But Sebastião de Melo's greatest reforms were economic and financial, with the creation of several companies and guilds to regulate every commercial activity. He demarcated the region for production of ' Port* *to ensure the wine's quality**, …* |
| **Target:** | Why did e Melo demarcate the region for production of Port? |
| **Supervised:** | Why did Sebastio de Melo demarcate Port? |
| **Advantage Actor-Critic:** | Why did Sebastio de Melo demarcate the region for production of Port? |

### 4.5.1 Comparison with Supervised training

We compare our proposed Advantage Actor-Critic training with the supervised training regarding corpus-level BLEU-4 scores obtained on the test set of the different datasets in Table 9. We observe a consistent trend of equal or higher corpus-level BLEU-4 scores on all the datasets using our proposed Advantage Actor-Critic training, which provides empirical proof of the benefit of our proposed training framework in providing broader corpus-level improvements. However, like supervised training, the Advantage Actor-Critic training also fails to get a non-zero BLEU-4 score on the Restaurants domain of the SubjQA dataset, which indicates the need for specific-targetted approaches for handling domain-specific datasets with subjective questions]-answer pairs.

We also show a few examples in Table 10 to illustrate how Advantage Actor-Critic training generates better questions than supervised training. Table 10 shows a few examples from the SQuAD, SQuADShifts, and SubjQA datasets, where we can see that the questions generated using Actor-Critic training are more similar to the ground-truth question than those generated using supervised training.

Table 11: Examples from SQuAD dataset where Advantage Actor-Critic training generates better questions than SCST in terms of surface-level similarity with the ground truth.

| Example 1 | |
|---|---|
| **Paragraph:** | *… America based actor Donald Moffat, whose roles include American Vice President Lyndon B. Johnson in the film The RightStuff, and fictional President Bennett in Clear and Present Danger, was born in Plymouth.* |
| **Target:** | In what film did Donald Moffat play President Bennett? |
| **SCST:** | In what film was the fictional President Bennett released? |
| **Advantage Actor-Critic:** | In what film did Donald Moffat play President Bennett? |
| **Example 2** | |
| **Paragraph:** | *…The dungeons are connected by a large overworld, across which Link can travel on foot; on his horse, Epona; or by teleporting.* |
| **Target:** | What is the name of Link's steed? |
| **SCST:** | What horse can Link travel on? |
| **Advantage Actor-Critic:** | What is the name of Link's horse? |
| **Example 3** | |
| **Paragraph:** | *… Sub-prime loans made by CRA-covered institutions constituted a 3% market share of LMI loans in 1998, but in the run-up to the crisis, fully 25% of all sub-prime lending occurred at CRA-covered institutions and another 25% of sub-prime loans had some connection with CRA. …* |
| **Target:** | What percent of sub-prime lending occurred at CRA-covered institutions in the run-up to the financial crisis? |
| **SCST:** | How much of all sub-prime lending occurred at CRA-covered institutions? |
| **Advantage Actor-Critic:** | How much of all sub-prime lending occurred at CRA-covered institutions in the run-up to the crisis? |

### 4.5.2 Comparison with SCST

We can see from Table 9 that the Advantage Actor-Critic training consistently outperforms SCST across the various question generation datasets, both general ones like SQuAD as well as domain-specific ones like SQuADShifts and SubjQA. Here, too, we observe a consistent trend of equal or higher corpus-level BLEU-4 score, which shows the benefit of our proposed lookahead rewards-driven Advantage Actor-Critic training. Here, too, we show a few examples to illustrate the quality of questions generated by the Advantage Actor-Critic training and compare it with the questions generated using SCST. Table 11 lists an example from each dataset to show how the Advantage Actor-Critic helps generate better-quality questions than existing frameworks.

Table 12: Corpus-level BLEU-4 scores obtained by the $T5_{small}$ and the $T5_{base}$ model after Supervised, SCST and Advantage Actor-Critic training. For the $T5_{small}$ model, due to the availability of results from previous experiments, we report the best result obtained using either softmax, Top-$K$, or Top-$p$ sampling. For the $T5_{base}$ model, we only train the model using softmax sampling, which shows that the Advantage Actor-Critic training, even with vanilla softmax sampling-based exploration, leads to a noticeable improvement in performance on both general as well as domain-specific datasets.

| Model | $T5_{small}$ $\approx 60M$ **parameters** | | | $T5_{base}$ $\approx 220M$ **parameters** | | |
|---|---|---|---|---|---|---|
| **Dataset** | **Supervised** | **SCST** | **Advantage Actor-Critic** | **Supervised** | **SCST** | **Advantage Actor-Critic** |
| **SQuADShifts** | | | | | | |
| **Amazon** | 16.02 | 16.13 | 16.23 | 18.33 | 18.41 | 18.66 |
| **Wiki** | 23.49 | 23.49 | 23.54 | 24.21 | 24.35 | 24.49 |
| **News** | 21.14 | 21.14 | 21.37 | 22.99 | 23.01 | 23.10 |
| **Reddit** | 15.20 | 15.29 | 15.39 | 17.52 | 17.54 | 17.59 |
| **SQuAD** | | | | | | |
| **Paragraph-Level** | 18.65 | 18.65 | 18.86 | 20.36 | 20.39 | 20.69 |
| **Sentence-Level** | 17.55 | 17.61 | 17.92 | 19.03 | 19.05 | 19.38 |

## 5 Robustness of Framework Concerning Architectural Modifications

We follow an architecture-agnostic approach concerning the architecture of the seq2seq model to ensure that the Advantage Actor-Critic framework is robust and can be easily used to train different seq2seq models regardless of their specific architectures. Therefore, to test the robustness of our proposed framework concerning architectural modifications, we train two different transformer-based seq2seq models using our Advantage Actor-Critic training framework and see if the Advantage Actor-Critic training can successfully improve the performance of both models. Accordingly, we train a separate $T5_{base}$ transformer model in addition to the $T5_{small}$ transformer already used in the previous experiments for testing the efficacy of the different design choices used in our framework. The $T5_{base}$ transformer model has $\approx 220$ million parameters and is nearly four times as big as the $T5_{small}$ transformer model, which has only $\approx 60$ million parameters.

We compare the corpus-level BLEU-4 scores obtained by both models after the Supervised, SCST and Advantage Actor-Critic training on one general dataset-SQuAD, and one domain-specific dataset- SQuADShifts, and report the results in Table 12. We can see that both models got the highest corpus-level scores using Advantage Actor-Critic training across all the datasets, which shows that our proposed Advantage Actor-Critic training framework can indeed improve the performance of different seq2seq models regardless of the architectural variations. This trend also shows that we can easily use our proposed Advantage Actor-Critic training framework to train very large seq2seq models like Dong et al. (2019), Qi et al. (2020), Ushio et al. (2022) which are the current state-of-the-art, and improve their performance noticeably.

## 6 Limitations of Advantage Actor-Critic Training Framework

Despite the various benefits offered in terms of addressing the issues observed in existing methods used for training seq2seq models for AQG, the proposed Advantage Actor-Critic training framework still suffers from the following limitations arising from the non-architectural design choice of using an n-gram similarity-based reward function:

- *Inability to identify semantic relatedness:* In this training framework, we use BLEU score-based lookahead rewards to drive the Actor-Critic training. Although the BLEU score-based lookahead reward function helps the Advantage Actor-Critic training significantly improve the corpus-level fluency score compared to the Supervised pre-training, the n-gram similarity-based metric has certain limitations. It has no sense of semanticity and underestimates the rewards for paraphrases. For example, the second and third examples of Table 13 show that question generated using the Advantage

Table 13: Examples illustrating limitations of n-gram similarity-based reward function. For reporting, we only show the sentence fragment with the answer (coloured in red) and the corresponding sentence-level BLEU-4 score.

| Example 1 | | |
|---|---|---|
| **Paragraph:** | *… The concept of liberation (nirvāṇa) —the goal of the Buddhist path—is closely related to overcoming ignorance (avidyā), a fundamental misunderstanding or mis-perception of the nature of reality. …* | |
| **Target:** | What is the goal of the Buddhist path? | |
| **Supervised:** | What concept is closely related to overcoming ignorance? | [67.99] |
| **SCST:** | What concept is closely related to overcoming ignorance? | [67.99] |
| **Advantage Actor-Critic:** | What is the goal of the Buddhist path? | [111.68] |
| Example 2 | | |
| **Paragraph:** | *… Since 1985 the eventually 354 MW SEGS CSP installation, in the Mojave Desert of California, is the largest solar power plant in the world. …* | |
| **Target:** | Where is the largest solar power plant in the world located? | |
| **Supervised:** | Where is the SEGS CSP plant located? | [38.44] |
| **SCST:** | Where is the SEGS CSP plant located? | [38.44] |
| **Advantage Actor-Critic:** | Where is the largest solar power plant in the world? | [89.75] |
| Example 3 | | |
| **Paragraph:** | *… Both Clarkson and Guarini made a musical film, From Justin to Kelly which was released in 2003 but was widely panned. …* | |
| **Target:** | What was the name of the film that the two finalists made together? | |
| **Supervised:** | What film did Guarini and Clarkson make? | [34.53] |
| **SCST:** | What film did Guarini and Clarkson make? | [34.53] |
| **Advantage Actor-Critic:** | What was the name of the film made by Guarini and Clarkson? | [57.82] |

Actor-Critic training gets a higher sentence-level BLEU-4 score despite being semantically similar to the questions generated using supervised learning and SCST.

- *Inefficiency single ground truth-driven reward computation:* This limitation is also related to using a surface-level metric like BLEU for calculating rewards. In the current approach, we compare the question generated by the Actor with a single ground-truth question to calculate the per-step lookahead rewards. However, this strategy is inefficient as it fails to recognize other potentially good questions. For example, the first and second examples of Table 13 show that the question generated using Advantage Actor-Critic training gets a higher sentence-level BLEU-4 score because of being structurally similar to the ground-truth question. However, the questions generated using supervised learning and SCST, which ask questions with a different but contextually relevant intent, get lower sentence-level scores. The above limitations suggest the need for a better rewarding strategy that can identify semantic relatedness and recognize the contextual relevance of the questions asked with different intents.

## 7 Related Work

This paper proposes an Advantage Actor-Critic training framework to train seq2seq models for AQG. In our work, we follow an architecture-agnostic approach concerning the exact architecture of the seq2seq model and focus more on leveraging ideas from existing reinforcement learning literature to improve the performance of seq2seq models trained for AQG. Accordingly, we use a simple transformer architecture-based seq2seq model and a $< hl >$ tag-based input pipeline, like the *BERT-HLSQG* model (Chan & Fan, 2019), to feed the input passage-answer text to the seq2seq model. However, before the advent of transformer (Vaswani et al., 2017)-based large language models, researchers mostly used the teacher-forcing algorithm for training RNN-based seq2seq models for AQG, which typically consisted of an LSTM (Hochreiter & Schmidhuber, 1997) or GRU (Cho et al., 2014)-based encoder and decoder components. Although the overall procedure for training these models was the same, they differed in terms of the model architecture and the nature of the input. For example, Du et al. (2017), Zhao et al. (2018), and Song et al. (2018) were all RNN-based seq2seq models, which used different input pipelines to feed the passage-answer text as input to the seq2seq model. While Du et al. (2017) only used the passage as the input, Zhao et al. (2018) and Song et al. (2018) used both the passage and the answer and generated refined input representations which helped generate better-quality questions. Zhao et al. (2018) used meta-tags to implicitly provide information about the answer span in the input passage, and Song et al. (2018) explicitly modelled the information between the answer and other contexts within the passage by encoding the passage and the answer separately using different encoders.

Subsequently, researchers started re-formulating the AQG task as a sequential MDP and used the SCST framework based on the REINFORCE with baseline method to learn an optimal policy $\pi^*$, which maximized some linguistic aspect-specific reward corresponding to the question generated autoregressively by the seq2seq model. Nema et al. (2019) was one of the first works which trained an LSTM-based seq2seq model in the SCST setup for generating questions from any arbitrary passage-answer pair. Chen et al. (2020) also used the SCST approach for training a graph neural network (GNN)-based novel graph-to-sequence (graph2seq) model for AQG, which consisted of a Bidirectional Gated Graph Neural Network based encoder and an LSTM-based decoder to generate questions from a passage-answer pair. Our work differs from these approaches because we use an Actor-Critic method-based training approach to learn an advantage function, which we use to aid the seq2seq models in learning an optimal policy for the AQG task.

Previous works have used Actor-Critic training for other sequence-generation tasks like machine translation and summarization. However, our proposed Advantage Actor-Critic training framework is among the first to leverage Actor-Critic methods for AQG and training seq2seq models to generate questions from input passage-answer texts. Bahdanau et al. (2017) was one of the first works to use Actor-Critic methods to train the seq2seq model for sequence generation tasks. Their proposed Actor-Critic training framework used a separate Critic network for training seq2seq models for sequence generation tasks like machine translation. However, the Critic in their work was a seq2seq model, conditioned on the ground truth, which learned to generate action-value estimates for every possible state-action pair combination. These action-value estimates from the Critic provided dense training signals, which aided the Actor model in the machine translation task.

It was very challenging to train the Critic model because the Critic had to generate action-value estimates for all the tokens the Actor could output from a given state. For example, Bahdanau et al. (2017) observed that the Critic often overestimated the action values, which they partially tried to address by including a separate term in the Critic's optimization objective that penalized the variance in the Critic's output and helped prevent the overestimation of action-values of rarely samples tokens. In contrast, the Critic model in our Advantage Actor-Critic training framework learns the state-value function and uses it to compute the advantage estimates using the GAE trick. The state-value function is much simpler than the action-value function learned in Bahdanau et al. (2017) and significantly simplifies the Actor-Critic training process for sequence generation tasks.

## 8 Conclusions and Future Work

In this work, we proposed an Advantage Actor-Critic training framework based on the A2C method to aid seq2seq models in the AQG task. In the proposed Advantage Actor-Critic training framework, we proceeded with an architecture-agnostic approach, focusing more on leveraging ideas from existing reinforcement learning literature to introduce an Actor-Critic method-based training strategy that improves the seq2seq models' performance by enhancing the generated questions' quality. In the proposed training framework, we used learnt reinforced signals from a separate Critic model to guide the Actor model in the question-generation training process. We used a state-value estimator as the Critic, which learns the state-value value function to help compute normalized advantage estimates, based on the GAE model, for every token output by the Actor and guide the Actor in the question-generation training. Since we compute the advantage estimates at every decoding step using a temporal-difference (TD)-method, it overcomes the high-variance and sparse training signal issue associated with the Monte Carlo return-based policy gradient methods like SCST. We discussed how our proposed Advantage Actor-Critic training framework addresses the exposure bias problem by allowing the Actor to generate questions autoregressively using tokens from the Actor's policy distribution. We also discussed how the lookahead rewards-driven strategy optimizes the Actor towards a non-differentiable metric like BLEU. In this work, we addressed the intractable action space bottleneck by proposing a multi-step training strategy and probability-based action pruning. We also refined the overall training process by using an off-policy approach for training the Critic, with an explicit division of high-reward and low-reward experiences. To study the efficacy of the proposed Advantage Actor-Critic training framework, we conducted multiple experiments on benchmark question-generation datasets from QG-Bench and thoroughly analyzed the proposed design choices. We also compared the Advantage Actor-Critic training with existing supervised learning and policy gradient-based training approaches and showed how our proposed training framework outperforms these methods by generating better-quality questions with a high degree of surface-level similarity with the ground truth.

Although we proposed supervised pre-training and action-pruning strategies to address the intractable action space bottleneck in our current work, we would like to investigate the viability of leveraging other methods from the reinforcement learning literature, like action-representation learning, to address this issue. Specifically, in order to address the problem of large action space inherent in natural language generation tasks, we can use methods like (Dulac-Arnold et al. (2015), Chandak et al. (2019)) to learn a multi-component policy composed of two separate sub-components, which are as follows:

- *Component 1*, which uses the information about the structure of the actions (tokens) to embed them in a low-dimensional continuous space of action representations.

- *Component 2*, which learns a separate function that maps the embedding from the low-dimensional continuous action space to a high-dimensional discrete action space.

Such a multi-component policy can potentially overcome the intractable action space bottleneck, and it would be interesting to see how it can help generate good-quality sequences for AQG.

Another exciting research direction is using semantic similarity-based reward functions instead of a structural similarity-based evaluation metric like BLEU to drive the AQG training. We would like to extend our current work by studying the use of different reward functions and how they affect the question-generation process

by improving the performance of the seq2seq model concerning different linguistic characteristics of the generated question as informed by the specific evaluation metric. In our current work, we discussed how a reward function based on an n-gram metric like BLEU incentivizes the Actor to generate questions with significant structural similarity with the ground truth and only improves the generated questions' quality concerning fluency. Such a reward function has no sense of semanticity, which causes it to fail in identifying paraphrased questions having semantic relatedness to the ground truth. Therefore, we would like to study other reward functions driven by semantic similarity-based metrics like BERTScore (Zhang et al., 2020) and analyze how such reward functions help improve the seq2seq models' robustness.

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
