# OpenReview forum: "Advantage Actor-Critic Training Framework Leveraging Lookahead Rewards for Automatic Question Generation"
_TMLR — Rejected by TMLR_

### Review · Reviewer_49Hm · 2023-06-05

**Summary Of Contributions:**

The paper discusses benefits arising from using an Actor-Critic framework for finetuning NLG architectures, in particular in the context of automatic question generation. The introduction is very interesting, discussing why such approaches, and more generally reinforcement learning, are useful for such purposes (for exposure bias, evaluation mismatch, etc) but face important challenges, such as delayed rewards (i.e., non nul reward only at the end of the sequence generation) and intractable action space. The global approach seems quite reasonable and results look good, while not very surprising as it follows mostly classical choices from the literature of the domain.

My main concern is that I cannot find what are the main innovations of the approach. While authors announce many contributions in the end of the introduction, none of them look to me as a really new proposal: 1) Actor-Critic: many NLG approaches using an actor-critic framework, notably RLHF approaches such as in chatGPT, use approaches like PPO (a central RL algorithm from the actor-critic family), or approaches like MALIGAN rely on a critic Q network [1]; 2) Lookahead reward: the proposal looks innovative but I cannot really understand what it brings compared to a true GAE n-steps returns with TD-lambda updates for instance; 3) Probability-based action pruning: This is borrowed from approaches like Nucleus (cited by authors), already used many times in the NLG litterature (see for instance [2],[3]); 4) Off-Policy Critic training with an explicit division of high and low reward samples: this could be an interesting contribution, but the paper misses comparisons with approaches such as for instance Prioritized Experience Replay, and discussions (including theoretical analysis) about why this is beneficial; 5) Multi-step Training: This is not a contribution, most of NLG approaches based on RL finetuning work like this.

The paper looks more like the technical description of a considered architecture than a research paper, despite an interesting related work section. My feeling is that authors should focus greatly more on one single contribution from their list, which they work further with theoretical analysis and at least greatly stronger empirical evaluation, comparing the proposal with other popular similar techniques from the litterature, which is not done here. We only have ablations of some choices, no real comparison with concurrent techniques.

[1] Che, T., Li, Y., Zhang, R., Hjelm, R. D., Li, W., Song, Y., & Bengio, Y. (2017). Maximum-likelihood augmented discrete generative adversarial networks. arXiv preprint arXiv:1702.07983.

[2] Sylvain Lamprier, Thomas Scialom, Antoine Chaffin, Vincent Claveau, Ewa Kijak, Jacopo Staiano, Benjamin Piwowarski:
Generative Cooperative Networks for Natural Language Generation. ICML 2022: 11891-11905

[3] 	Thomas Scialom, Paul-Alexis Dray, Sylvain Lamprier, Benjamin Piwowarski, Jacopo Staiano:
ColdGANs: Taming Language GANs with Cautious Sampling Strategies. NeurIPS 2020



**Audience:**

No

**Broader Impact Concerns:**

.

**Claims And Evidence:**

No

**Requested Changes:**

I would suggest to focus on either lookahead reward or training dataset division, with stronger analysis, to make their paper useful for the research community.

Also I cannot see why authors restrain the scope of their approach to automatic question generation, while the proposal does not include any specific feature from that task and could/should be studied in a broader context. The positioning from the state of the art stating that Actor-Critic frameworks have already been proposed for NLG in the litterature, but not for question generation, is very weak (nothing specific for AQG here) and not fully true.

Section 6 about limitations discusses well known limitations about finetuning language models with ground truth metrics such as BLEU. Many alternatives have been considered in the litterature (e.g. based on learned discriminator, density ratio estimators, external question-answering systems, etc.): why authors still work with this here ?

Big architectures pictures from fig 2 to 6 are redondant and unclear. I would suggest to remove all of them (maybe keep one overview but more dense with more insightful components) and rather give mathematical formulations of the architectures.

Notation s_t from fig 1 is very confusing in RL point of view, as it is not the state but a learned representation of it. should be replaced by something like \phi(s_t) or something like this.

Please give more details of the mathematical components of formulas. For instance A^GAE_\psi is not defined, while it does not look as the classical GAE advantage as defined in the corresponding paper which uses TD-lambda updates. This is very important as if TD-lambda would be used, the interest of the lookahead reward definition would look redondant. I cannot see why t would be better than using TD-lambda for giving dense targets to the critic network. Also, the critic is classicaly used to predict the average future reward. This does not look to be the case here. I suspect that the proposal modifies the optimal policy, which should be at least discussed in the paper.

For me the direct  use of Top-p Sampling inside an Actoc-Critic framework leads to a biased policy gradient, as it does not follow the learned policy. See for instance in [3] for Nucleus use inside mixtures and importance sampling to avoid such biases.


**Strengths And Weaknesses:**

Strengths:
   - Interesting introduction well setting current challenges in NLG
   - Practical architecture with Interesting results

Weakness:
    - Not very innovative
    - Proposals not enough justified and studied
    - weak positioning w.r.t. state of the art

---

> ### Author Response · Authors · 2023-07-29
> **Response to Reviewer 49Hm**
>
> We want to thank the reviewer for the valuable feedback. We want to take this opportunity to address some of the concerns raised by the reviewer:
> * _Restriction of the scope of the proposed training framework to Automatic Question Generation:_ We mentioned that the primary objective was to leverage ideas from existing literature to provide a stable learning framework for addressing the four specific issues present in the existing QG strategies: exposure bias, evaluation measure mismatch, global reward problem and intractable action space bottleneck _(Section 1)_. As the reviewer suggested, these problems are not restricted to question generation but are also present in other NLG tasks. However, despite being interesting, the analysis of the effect of the proposed training framework in the context of a wide range of other text generation tasks like machine translation or summarization was simply beyond the scope of the current study due to specific nuances involved in these different NLG tasks.
>
> * _Use of BLEU-4 evaluation metric to compute rewards:_ Given that our primary objective was to propose a stable architecture-agnostic Actor-Critic method-based training framework to improve the quality of the seq2seq model's output concerning any specific linguistic aspect, we limited our focus in the current study to simply improving the fluency of the generated questions _(Section 1)_. Prior works have defined fluency in terms of the BLEU metric, which also happens to be the primary evaluation metric in most existing works in the question generation literature. Therefore, we decided to proceed with the BLEU metric as the reward function of choice in our current work. However, we also tried to ensure the readers realize the downside of this approach by specifying the limitations of using such a structural similarity-based reward function _(Section 6)_.
>
> * _Lack of details of the mathematical components of formulas:_ We believe we explained the mathematical components and cited the related works in the relevant sections of the paper. For example, we have mentioned in _Section 2.4_ that $A^{GAE}_{\psi}$ is the normalized advantage score estimated using the Generalized Advantage Estimation trick proposed by _[1]_.
>
> **Response to Other Requested Changes**
> * _Replacement of Fig 2 to Fig 6 with a single detailed figure:_ We thank the reviewer for the valuable suggestion. Keeping the suggestion in mind, we plan to replace these figures with a single comprehensive figure in the revised version of the paper. However, we feel that Fig 5 and 6 are essential as they help give the readers an overview of the Actor and Critic training steps, and we plan to retain these with some minor modifications related to the orientation of the text.
>
>
> **Cited papers**
> * _[1]:_ John Schulman, Philipp Moritz, Sergey Levine, Michael I. Jordan, and Pieter Abbeel. High-dimensional continuous control using generalized advantage estimation. In Yoshua Bengio and Yann LeCun (eds.), 4th International Conference on Learning Representations, ICLR 2016, San Juan, Puerto Rico, May 2-4, 2016, Conference Track Proceedings, 2016. URL http://arxiv.org/abs/1506.02438.

---

### Review · Reviewer_3dvU · 2023-06-20

**Summary Of Contributions:**

This paper considers the task of generating questions (AQG) about a given piece of text and a candidate answer (to the generated question):
1. They propose to train seq2seq text generation models using an RL framework. They depart from traditional approaches for using RL to train seq2seq models by using a critic model to provide the seq2seq model (the "actor") reward signal at each generation step).
2. They provide this signal by training the critic with a look-ahead mechanism so that it can predict the global, sentence-level reward.
3. They also train the critic using a prioritized replay buffer for off-policy critic training.
They run experiments using question generation datasets like SQuAD and verify that the proposed changes do indeed produce an improved AQG model. They run some ablations of their design decisions.

Other Notes
* S1: Introduction
    * “The neural networks are sequence-to-sequence (seq2seq) models composed of an encoder and a decoder.”: NNs aren’t always seq2seq and even for question generation, they don’t have to be.
    * “one crucial difference between AQG and such sequence generation tasks is that we can generate multiple questions from the same content depending on the question’s intent.”: This also is true of text summarization
    * The examples in Table 1 seem fairly cherry picked
        * How were the examples selected?
        * How are the model outputs generated?
        * It’s not clear to me that the outputs of advantage actor-critic are better: There is much more word overlap with the original paragraph, which could be bad in several situations (generating training data, trying to produce hard reading comprehension questions, etc.)
    * The problem statement is really a problem statement (it is presenting a solution). I recommend just cutting it; the contribution section is clear enough in what the paper is doing.
    * One of the stated contributions is “Probability-based action pruning for efficient exploration of vocabulary space” but really they mean top-p and top-k decoding, which has been well known in the NLG community for several years.
* S2
    * It’s odd to frame AQG as finding the argmax and ignore the search problem (over sequences) when the next section is framing the search problem as an MDP
    * I’m not sure what Figures 3 and 4 are telling me that is different from Figure 2
    * Figures 5 and 6 are hard to read with the text orientations
        * I would much prefer figures that are oriented the same way as the rest of the paper
    * 2.5: moving from deterministic decoding algorithms like beam search to probabilistic ones is a pretty significant change. In some cases, probabilistic decoding may not be acceptable (e.g. for reproducibility reasons).
* S3
    * Should give more details on the models being used from Ushio et al. (2022): What were they pretrained on? Are they just T5 models or T5 models fine-tuned specifically for AQG? What are the relevant training details (possibly in an appendix)?
* S4
    * 4.3: Seems like an odd question to investigate given that your main experiments are generating with beam search
    * 4.5
        * Better to combine Tables 3 and 9 (and their discussion) as they both are comparing the core modeling changes.
        * You should provide (at least a brief) description of methods you are comparing against, e.g. SCST
* S5
    * It seems a bit silly to say that switching to T5 base from T5 small is a substantive architecture modifications (though they are fairly different in parameter count)
    * “This trend also shows that we can easily use our proposed Advantage Actor-Critic
    * training framework to train very large seq2seq models”: that’s a pretty strong conclusion to draw; 220M parameters is a world of different from tens to hundreds of billions of parameters
* S7
    * “However, before the advent of transformer-based large language models, researchers mostly used the teacher-forcing algorithm for training RNN-based seq2seq models for AQG, which typically consisted of an LSTM”: quite a few works still used Transformer based seq2seq models (e.g. T5); it seems odd to draw the line at Transformers


**Audience:**

Yes

**Broader Impact Concerns:**

no concerns

**Claims And Evidence:**

No

**Requested Changes:**

* [critical] Present an experiment and/or evaluation demonstrating that the generated questions are indeed improved, either through using human evaluation or a high-quality learned metric/model.
* [critical] Use a reward function that is better than BLEU at capturing question quality.
* [critical] Reword the contribution "Probability-based action pruning for efficient exploration of vocabulary space" as it is currently worded to suggest that you proposed top-p/k decoding, which is not accurate, and make more clear what your contribution here is.
* Some of the figures seem highly redundant and are hard to read (e.g. are oriented vertically with horizontal text boxes). These figures should be consolidated with clearer captions and layouts. See "Other Notes" in "Summary of Contributions"
* Typos
    * “Reinforce” —> “REINFORCE”
    * “Rl-based” —> “RL-based”
    * “Automatic QUestion Generation”: s2.1 title
    * “Inefficiency single ground truth-driven reward computation” —> “Inefficient single …”
* Missing citations
    * Cite REINFORCE
    * Cite BLEU
    * “A Deep Reinforced Model for Abstractive Summarization” (Paulus et al. 2017): an early and influential work on using RL + ROUGE to train seq2seq text generation models

**Strengths And Weaknesses:**

A strength of this work is that the author has identified some legitimate problems and bottlenecks when using RL (e.g. REINFORCE) to train NLG systems. The proposed changes directly address these limitations and theoretically solve some of these issues.

I see a few major limitations and weaknesses of this work
1. The evaluation metric (BLEU-4) is the same used to compute rewards when training the model in the new framework. The experiments show that BLEU-4 goes up with this training, but this is not surprising given that you are directly optimizing for this metric. I am not convinced that the resulting models are actually better at generating questions, as there is no other attempt at evaluating the quality of the generations other than cherry picking some model generations.
    a. From the examples shown in the tables, I'm actually concerned that the generated questions are worse because they more closely match the surface form of the supporting text (and the ground truth question). For some applications, this high degree of word overlap is bad because it allows for the questions to be answered more easily (e.g. for augmenting QA datasets or generating exams to test reading comprehension).
2. Relatedly, previous work, e.g. “A Deep Reinforced Model for Abstractive Summarization” (Paulus et al. 2017), has identified that using RL-based optimization to n-gram based metrics (e.g. ROUGE) actually leads to worse and less fluent outputs. This work should attempt to explore and verify whether the generated questions see a similar degradation in a rigorous, quantitative experiment that is not based on using BLEU to evaluate.
3. BLEU is an imperfect and even poor metric for evaluating the quality of generated questions. The authors admit as much in the limitation section (S5), but do not explore using alternative metrics (e.g. a learned metric, like BERTScore or a preference model) to train their models. I don't think this is a fatal flaw in the paper, as in theory it is easy to swap BLEU for another metric, but ideally the paper would be expanded to use a more robust and trustworthy reward signal.
4. A more fundamental concern I have is the per-token rewards, which are computed using the generation history / prefix of questions generated so far. BLEU on prefixes definitely does not seem to give unilaterally correct rewards. However, it's not clear to me that you can design a clear or correct reward for a prefix because of the large number of possible ways of wording a question (e.g. "How does backprop work?" versus "Describe backprop"). At a minimum, the early rewards would be highly uninformative. Given the large space of acceptable questions, it is worth investigating whether or not the per-token rewards are useful (moreso than BLEU evaluation can tell you).

---

> ### Author Response · Authors · 2023-07-29
> **Response to Reviewer 3dvU**
>
> We thank the reviewer for their valuable observations and for pointing out the typos. We want to address some of the concerns raised by the reviewer below:
> * _Use of BLEU-4 evaluation metric to compute rewards:_ In our current work, our primary objective was to address the issues present in the existing QG strategies: exposure bias, evaluation measure mismatch, global reward problem and intractable action space bottleneck _(Section 1)_. We decided to proceed with the BLEU metric as the reward function of choice because most prior works in the existing literature have the BLEU score as the primary evaluation metric. However, we ensure the readers realize the downside of this approach and consequently mention the limitations of using such a structural similarity-based reward function _(Section 6)_, as stated by the reviewer.
>
> * _Lack of exploration of other evaluation metrics apart from BLEU:_ As specified in our previous response, our primary objective in this study was to address the issues present in the existing QG strategies _(Section 1)_. Therefore, we limited our focus to providing a stable learning framework to solve these problems. Consequently, we only target the linguistic aspect of fluency in our current study and use the BLEU metric as our reward function's basis. The investigation of the effect of other reward functions, especially those driven by semantic similarity-based metrics like BERTScore, which can serve as a possible solution to mitigate issues arising from the use of a BLEU metric-based reward function, is left as a part of future work (Section 5).
>
> * _Lack of verification of the model's tendency to generate degenerate questions:_ We tried to address some of the degeneracy in the output text, similar to the ones observed in the work cited by the reviewer. For example, we use an explicit n-gram penalty mechanism, similar to the one suggested in the cited work, to prevent the occurrence of repetitive bi-grams in our generated questions _(Section 3.2)_. However, keeping the reviewer's valuable suggestion in mind, we plan to include a separate section to discuss the model performance on additional metrics, which can provide a more comprehensive view of the quality of generated questions.
>
> * _BLEU metric's inability to correctly reward question prefixes:_ We mentioned how the BLEU metric, even after the application of smoothing, is still a sentence-level metric, which we cannot use directly to generate rewards for intermediate question prefixes (Section 2.3). However, we also mentioned that we can still use the sentence-level BLEU score to derive latent information about the quality of intermediate prefixes which compose the final question. Further, to account for the many possible ways of wording a question, we introduced the $\beta$ parameter, which specifies the model's confidence about the final sentence-level score at any given step. Having a $\beta < 1$ means that the model is not entirely sure about the final sentence-level score at the intermediate steps, which, we believe, indirectly accounts for the possibility of different possible questions.
>
> **Response to Other Requested Changes**
> * _Reword the contribution "Probability-based action pruning for efficient exploration of vocabulary space":_ We have already cited the source material for the Top-K and Top-p sampling, which we propose leveraging as action pruning approaches to reduce the vocabulary-based action space _(Section 2.5.3)_. However, we can rephrase the sentence if required to remove any possible ambiguity.
>
> * _Modification of figures to remove redundancy and improve clarity:_ We thank the reviewer for this valuable suggestion and plan to replace Fig 2 to Fig 4 with a single comprehensive figure and correct the orientation of the text in Fig 5 and 6 to improve the clarity.
>
> * _Typos and missing citations:_ We will correct the typos and include the missing citations in the relevant places.

---

### Review · Reviewer_QeRU · 2023-07-12

**Summary Of Contributions:**

This paper proposes a few modifications to baselines in order to achieve better performance on automatic question generation through reinforcement learning. In particular, it uses an actor-critic optimization algorithm as opposed to prior works that use policy gradients, it also densifies the reward by scaling the global reward at each step (token) of generation. Additionally, it is proposed to use top-k or top-p sampling in order to shrink the action space and make RL more tractable.

**Audience:**

Yes

**Broader Impact Concerns:**

There is no broader impact statement, but I do not believe this work in general exacerbates the issues with language modeling that already exist. In particular, question generation is very targeted, so this is not a barrier to acceptance.

**Claims And Evidence:**

No

**Requested Changes:**

- I would like to see an experiment that uses policy gradients (SCST) with the proposed dense reward. This would be critical to my decision (or an explanation if there is some misunderstanding in Weaknesses).
- Some more discussion of related work using RL and sequence modeling is necessary. The final two paragraphs of the related work are dominated by a single citation from 2017, and it is hard for me to imagine that nothing has taken place in the meantime. This is critical to my decision.
- I think it is important to conduct either a human evaluation, or to evaluate with a different and orthogonal automatic metric so as to measure performance along dimensions that the method is not directly optimized for. The choice of BLEU score for both training and evaluation creates a tighter overlap for the RL methods than for supervised learning, which may make the results appear artificially stronger. I recognize that this may be a tall order, but I would appreciate even a small sample size of raters. Some evaluation like this would be critical to my decision.
- It would be interesting to see some sort of qualitative evaluation of the diversity of questions that this method results in, when compared to other baselines, as that is the proposed rationale in AQG in the introduction, this is not critical to my decision.
- I would prefer the related work come earlier in the document, after the introduction, this is not critical but makes for a better read.

**Strengths And Weaknesses:**

Strengths:
- The gains over baselines are strong.
- All components of the method are clearly ablated
- The description is very clear, and thus implementable from the text alone.
- The writing in general is high quality.

Weaknesses:
- The proposed method explicitly optimizes for BLEU score, and the main evaluation metric throughout the text is also BLEU score. Such a tight overlap between the RL methods training objective and evaluation may inflate the performance against supervised learning when considering the true metric we care about: human judgment. As a result, it would be nice to see an evaluation using some other metric that validates the performance of the method.
- As far as I understand, the primary baseline SCST that is policy-gradient based does not use a densified reward. As such, it is difficult for me to understand the importance of the actor-critic off-policy setup in the experiments. Is the gain primarily due to the denser reward?
- Though the method is motivated throughout the paper, it seems to me that this motivation is not specific to AQG, but rather to language modeling as a whole. The primary difference highlighted between AQG and other tasks is that AQG allows for multiple questions to be generated from the same content. I'm not sure that this is not true of summarization as claimed, but granting that claim it is not clear to me that this particular aspect of AQG is explored. There is not explicit empirical validation that the learned distribution of questions is more diverse, only general performance improvements.
- Though it is claimed that the framework is general (and I would believe that it should be), it is only tested on two models of the same architectural family (T5-small and T5-base). In particular, I do not consider T5-base to be a significant "architectural variation" from T5-small.
- The overall method involves many moving parts, and given that it is only applied to AQG, it is unclear the particular match of these moving parts to the task.
- Related work comes at the end of the paper. As a reader somewhat unfamiliar with more recent prior work, it makes it difficult to contextualize the contributions of the paper, though I should note it is strange to me that many citations in the same domain are somewhat old by the field's standards, and the baseline compared to dates from 2016.

---

> ### Author Response · Authors · 2023-07-29
> **Response to Reviewer QeRU**
>
> We thank the reviewer for their valuable observations. We would also try to address some of the concerns of the reviewer below:
>
> * _Inflation of model's performance due to using BLEU-4 as training objective and evaluation metric:_ Our primary objective was to provide a stable learning framework to solve these issues, and we limited our focus to improving the specific linguistic aspect of fluency, indicated by the BLEU metric. We did not investigate the analysis of the effect of different evaluation metrics-based reward functions on the model's overall performance as it was outside the primary objective and the scope of the current study. However, we tried to drive the reader's caution towards the downsides of using the BLEU metric-based reward function _(Section 6)_. However, keeping the reviewer's valuable suggestion in mind, we plan to include a separate section to discuss the model performance on additional metrics, which can provide a more comprehensive view of the quality of generated questions.
>
> * _Lack of analysis of SCST with dense lookahead rewards:_ We did not analyze the effect of dense lookahead rewards-based SCST on the model's performance because our focus was more towards comparing and contrasting our proposed Advantage Actor-Critic framework against existing strategies widely used in the Automatic Question Generation literature _(Section 5)_. Since current SCST strategies typically use sentence-level scores, we did not explicitly conduct dense lookahead rewards-based SCST experiments. However, if required, we can include the said results in the revised version of the paper.
>
> * _More discussion of related work using RL and sequence modeling:_ Currently, we limited the discussion of RL and sequence modeling in the related work section to a few works which leveraged SCST for AQG and Actor-Critic methods for other text-generation tasks and tried to provide a picture of how our work compares to them. In the case of SCST, we cited papers from 2019 and 2020, while in the case of Actor-Critic methods, we primarily discussed work from 2017. There have indeed been other recent works that have leveraged Actor-Critic methods for orthogonal text-generation tasks. However, we did not include those because we wanted to primarily focus on related advancements in the question-generation literature, in which supervised learning-based approaches and a few policy gradient-based ones have relatively dominated. However, keeping the reviewer's suggestion, we plan to include a section to discuss more recent works related to the application of Actor-critic methods in language modeling tasks.
>
> * _Inadequate examples to prove the proposed framework's robustness concerning architectural modifications:_ In our proposed Advantage Actor-Critic training framework, we ensured that all of its working components were independent of the architecture of the seq2seq model-based Actor. We verified this using two readily available $T5$ transformer models from _[1]_ _(Section 3.1)_, released with the QG-Bench benchmark. These models are already trained in the supervised learning setup allowing us to readily proceed with the reinforcement learning part of our proposed training framework. Our primary objective in these tests was to see if our framework successfully improves the performance of both small-sized and large-sized seq2seq models. We did not feel the need to explicitly repeat the experiments with other types of seq2seq models (e.g., smaller RNN-based models or other much larger transformer-based models) because we felt the results of the current experiments with the $T5_{small}$ and $T5_{base}$ transformer models were enough to prove the capability of the training framework to improve the performance of models concerning any specific linguistic aspect.
>
> **Response to Other Requested Changes**
> * _Lack of human evaluation or evaluation with other orthogonal metrics:_ Although the human evaluation experiment would be expensive and challenging due to time constraints, we plan to include a separate section to discuss the model performance on additional metrics, which can provide a more comprehensive view of the quality of generated questions, something we already mentioned in our first response.
>
> * _Placement of the "Related Works" section earlier in the document:_ We thank the reviewer for this valuable suggestion and plan to implement the same in the revised version of the paper.
>
> **Cited papers**
> * _[1]:_ Asahi Ushio, Fernando Alva-Manchego, and Jose Camacho-Collados. Generative language models for paragraph-level question generation. In Proceedings of the 2022 Conference on Empirical Methods in Natural Language Processing, pp. 670–688, Abu Dhabi, United Arab Emirates, December 2022. Association for Computational Linguistics. URL https://aclanthology.org/2022.emnlp-main.42.

---

> > ### Comment · Reviewer_QeRU · 2023-08-08
> >
> > - **Inflation of model's performance...**: I'm happy to see the authors plan, but it is unclear to me which additional metrics they plan to evaluate on. In particular I would be happy with some metric for diversity of questions generated (in keeping with the theme of AQG allowing for diverse generations), as well as an orthogonal measure of quality besides BLEU. I believe a small-scale human evaluation is not too much to ask for in a short rebuttal period, but ultimately I'd be satisfied with the two.
> > - **Lack of analysis...**: I mentioned this as it would shed more light on where the source of improvements are coming from compared to baselines. I believe this should be included in the main text, especially as it seems it is not too much burden for the authors.
> > - **More discussion of related work...**: Still, it is possible that many recent advances outside the AQG literature would be applicable, and would constitute stronger baselines than those presented in the text if implemented by the authors. I'm not tied to this point, but I don't think it is good practice to scope the competing methods so narrowly, especially if there are well-known methods that use similar tools on different domains than AQG.
> > - **Inadequate examples...**: I agree that RNNs are as of this current moment worse than existing Transformers. While I understand the authors' intent, perhaps it is better to rephrase the contribution as applicable across model-scales, rather than across architectures. The only thing that changes between T5-small and base is the size.
> >
> > I think with the changes discussed above I would be happy to recommend accept, though it is not clear to me how to verify them at this moment.

---

### Comment · Action_Editors · 2023-07-27
**Author Response to Reviews**

The manuscript has received three reviews and the reviewers agree that the paper nicely identifies several limitations that arise when training natural language generation models using reinforcement learning. They find the proposed architecture to be clearly presented, which makes it practical to implement, and that the results reveal that the model addresses several of these challenges.

The reviewers additionally raise several issues with the paper, perhaps most notably questions about the relevance of using BLEU score as the evaluation metric and for per-token rewards.

It would be helpful if the authors could respond to these reviews as soon as possible, as the reviewers will soon be submitting their final recommendations.

---

### Decision · Action_Editors · 2023-10-05

**Recommendation:** Reject

**Comment:**

The paper describes an integrated framework that seeks to improve the performance of automatic question generation (AQG) via reinforcement learning (RL). Underlying the framework is the use of actor-critic-based optimization as an alternative to policy gradients, as well as per-token densification of the global reward. Experiments demonstrate that the framework achieves strong performance gains relative to baselines.

The paper was evaluated by three reviewers who largely agree on the strengths and weaknesses of the paper. Among the strengths, the paper does a nice job identifying legitimate problems that arise when using RL to train language generation models. The paper provides a clear description of the components that are proposed to address these challenges, making it possible for others to implement the framework based on the text alone.

However, the reviewers agree on the core weaknesses of the paper in its current form that make its contributions unclear. First, the paper does not adequately situate this work in the context of existing approaches that use reinforcement learning for natural language generation (NLG), including those that employ actor-critic methods. While this may have been a deliberate effort to focus the discussion on the specific domain of automatic question generation, the reviewers and AE find the scope to be overly narrow, calling into question some of the paper's claimed contributions. Second, the reviewers agree that a significant weakness of the paper is that it is trained and evaluated using the same automatic n-gram metric (BLEU), which raises important concerns. Among them, it obfuscates the actual performance of the method as it is not surprising that it achieves gains in BLEU score when it is specifically trained to maximize this metric. As a result, this approach inflates performance gains over the baselines. Further, it does not provide sufficient evidence of the method's ability to generate high-quality questions, in part due to the well known limitations of BLEU-based metrics, which the paper acknowledges, as well as due to existing work that shows that using RL to train an NLG system using similar n-gram-based reward (e.g., ROUGE) actually results in worse, less fluent outputs (see the reference provided by Reviewer 3dvU). The authors offer to provide additional results that include different metrics, though they do not provide details regarding which metrics and evaluations would be added. Third, the reviewers question the use of per-token-based reward densification, specifically whether per-token rewards improve performance as noted by Reviewer 3dvU. This requires a proper ablation of per-token rewards and a comparison to an SCST baseline that similarly includes reward densification as pointed out by Reviewer QeRU.

The reviewers acknowledge the paper's potential, but agree that the aforementioned issues, which the authors' responses did not adequately address, limit the contributions of the paper in its current form.

**Audience:**

The application of reinforcement learning to automatic question generation and, more generally to natural language generation, is of interest to many in the ML community.

**Claims And Evidence:**

Several of the claimed contributions of the paper are inaccurate in light of existing work, including in the context of actor-critic-based approaches to natural language generation, which are not adequately discussed.

**Resubmission Of Major Revision:**

The authors may consider submitting a major revision at a later time.